# Gaming Consensus:
# Coordinated Manipulation in Crowdsourced Fact-Checking

**Nikil Roashan Selvam** [1]  **Jay Baxter** [2]  **Sophie Hilgard** [2]  **Brad Miller** [2]  **Keith Coleman** [2]  **Ellen Vitercik** [1]
**Sanmi Koyejo** [1]

## Abstract

*Crowdsourced fact-checking systems* have been adopted by major social media companies such as X, Meta, TikTok and Google with the aim of combating misleading information at scale without relying on centralized editorial control. These systems have been developed around a common underlying concept: a *bridging mechanism* that identifies notes flagging misleading information when they receive support from people with different perspectives rather than simple majority support. To our knowledge the only publicly disclosed bridging algorithms deployed for fact-checking are based on matrix factorization, as deployed by both X and Meta, augmented with additional components addressing abuse, targeted manipulation, and contributor brigades. This work examines the core matrix factorization portion of these systems, presenting theoretical and empirical evaluations of the degree to which coordinated users could vote strategically by leveraging the latent representations to fabricate the appearance of *synthetic consensus* within the bridging mechanism. Using historic production data, we find that up to 10.7% of lower quality notes could be manipulated above consensus thresholds using less than 10 ratings. We complement these findings with a theoretical analysis, revealing counterintuitively that rating a note as "Not Helpful" can increase its helpfulness score, as well as a cost model quantifying manipulation effort. We have developed and deployed mitigations within X's Community Notes algorithm to address synthetic consensus.

## 1. Introduction

Community Notes (Wojcik et al., 2022) is a crowdsourced fact-checking system pioneered by X,[1] with similar systems now used by Meta,[2] TikTok,[3] and Google,[4] to combat misinformation at scale without relying on centralized editorial control. When a post is potentially misleading, these systems typically allow a user to propose a note to provide additional context. These notes often consist of a 1-2 sentence explanation of why the post may be misleading, accompanied by links to supporting sources such as news articles, official statements, or datasets.

Once proposed, notes enter a rating phase where contributors evaluate them on whether the note is helpful and whether it provides accurate context. Crucially, a note does not become publicly visible simply because many users rate it as helpful. Instead, Community Notes[5] uses a bridging algorithm to only surface notes that achieve *diverse agreement*: that is, support from users who tend to disagree with each other on other notes. This design choice is intended to increase the likelihood that displayed notes are found helpful to people from different points of view. The bridging algorithm, as deployed by companies such as X and Meta,[6] is based on matrix factorization and complemented by additional components specifically addressing abuse, targeted manipulation, and contributor rating brigades.

Since 2022, X Community Notes has been globally available at scale, including to motivated and well-resourced adversaries. In that time, results have shown that X Community Notes effectively reduces misinformation spread (Chuai et al., 2024; Slaughter et al., 2025), amplifies fact-checks (Borenstein et al., 2025) and increases user trust in fact-checking (Drolsbach et al., 2024). Additional

---

[1] https://communitynotes.x.com/guide/en/about/introduction
[2] https://transparency.meta.com/features/community-notes
[3] https://newsroom.tiktok.com/en-us/footnotes
[4] https://blog.youtube/news-and-events/new-ways-to-offer-viewers-more-context/
[5] Unless specified otherwise, we use the term "Community Notes" to broadly denote crowd-sourced fact-checking systems rather than any particular company's production implementation.
[6] https://about.fb.com/news/2025/03/testing-begins-community-notes-facebook-instagram-threads/

[1] Stanford University, Stanford, CA, USA [2] X Community Notes, xAI, Palo Alto, USA. Correspondence to: Selvam et. al. <icml.consensus@gmail.com>.

*Proceedings of the 43rd International Conference on Machine Learning*, Seoul, South Korea. PMLR 306, 2026. Copyright 2026 by the author(s).

work has explored technical advances such as improved consensus mechanisms (De et al., 2025) and integration of large language models (Li et al., 2025; Wu et al., 2026). Adoption has correspondingly grown, with Meta eliminating third-party fact checks for US users.[7]

In this work, we develop the first rigorous analysis of the resistance of the core matrix factorization component of Community Notes to coordinated manipulation. Bridging-based algorithms (Ovadya & Thorburn, 2023; Small et al., 2021) offer some inherent resistance to manipulation: unlike traditional voting systems where adversaries can fabricate volume through fake upvotes, bridging requires *diverse agreement* across users with opposing viewpoints. Forging bridging consensus is harder as it requires an understanding of users and rating patterns on the platform. However, we demonstrate that it is possible to create *synthetic consensus* through a carefully designed voting strategy. Our approach does not rely on exploiting any software/security vulnerability or implementation bug, instead leveraging a fundamental property of the system's core matrix factorization algorithm.

This analysis is based on the open data and source code of X Community Notes, which facilitates independent study, critique, and improvements like those described here.

**Contributions.** We present a novel adversarial attack on the matrix factorization component of Community Notes, demonstrating that coordinated groups can manipulate the system to strategically fabricate diverse agreement. Our contributions include:

- A two-phase attack strategy where adversarial accounts first establish diverse positions in latent factor space, then coordinate to boost a target note's helpfulness score according to the matrix factorization algorithm.

- Empirical analysis on production data showing up to 10.7% of note quality scores below the median based on at least 10 ratings could be manipulated above the consensus threshold using less than 10 coordinated ratings.

- A theoretical analysis revealing a counterintuitive property: in some cases, rating a note "Not Helpful" can increase its helpfulness score.

- A cost model quantifying manipulation effort, which motivates the deployed mitigations in X's Community Notes system.

**Conflict of Interest Disclosure** The authors JB, KC, SH and BM are employed by xAI, where they work on X Community Notes. NRS was previously employed as an intern at xAI on the X Community Notes team.

[7] https://about.fb.com/news/2025/01/meta-more-speech-fewer-mistakes/

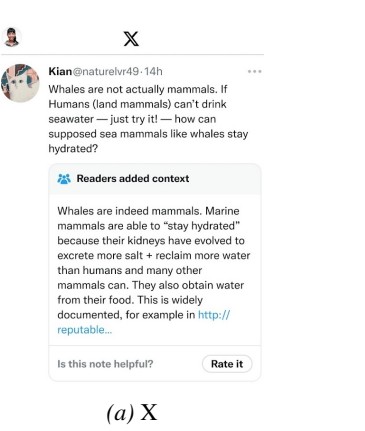

*(a)* X

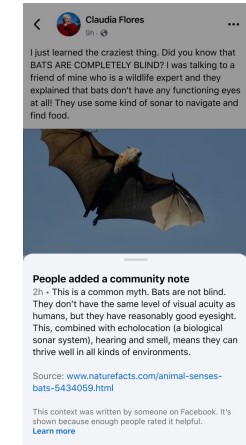

*(b)* Meta

*Figure 1.* Examples of Community Notes across platforms. Originally deployed on X, they have now been adopted by Meta, TikTok, and Google to combat misinformation.

## 2. How Community Notes Works

Community Notes systems commonly depend on a matrix factorization algorithm (Koren et al., 2009; Srebro & Jaakkola, 2003; Srebro et al., 2004; Mnih & Salakhutdinov, 2007) adapted from collaborative filtering in recommender systems. This technique, famously popularized by the Netflix Prize competition (Bennett & Lanning, 2007), learns low-dimensional representations of users and items from sparse rating data. In the classic recommender setting, the goal is to predict a user's rating of a movie based on how that user and other users have rated other movies.

Community Notes repurposes this framework for a different task: identifying consensus across different ideological groups. The algorithm embeds both users and notes in a shared space such that ratings are approximated by the inner product between the learned embeddings of the user and the note. The formal definition of the algorithm follows below.

Although matrix factorization is a common core component in Community Notes and the focus of this paper, specific products can include mechanisms that are out of scope for this analysis and are designed to increase robustness. For example, X[8] includes co-rater clique detection, predictive flip modeling, status stabilization periods designed to gather more ratings for provisionally helpful notes, and others.

### 2.1. The Matrix Factorization Algorithm

Formally, let $r_{un} \in \{0, 0.5, 1\}$ denote the rating that user $u$ gives to note $n$, where $1$ indicates "helpful", $0.5$ indicates "somewhat helpful", and $0$ indicates "not helpful." The

[8] https://communitynotes.x.com/guide/en/under-the-hood/ranking-notes

algorithm models each observed rating as:

$$\hat{r}_{un} = \mu + i_u + i_n + f_u \cdot f_n \qquad (1)$$

where:

- $\mu \in \mathbb{R}$ is the global intercept, representing the average helpfulness rating across all user-note pairs.
- $i_u \in \mathbb{R}$ is user $u$'s intercept, capturing that user's tendency to rate notes as helpful or not helpful relative to the global average.
- $i_n \in \mathbb{R}$ is note $n$'s intercept, representing the note's overall quality or consensus helpfulness. A high $i_n$ indicates broad agreement that the note is helpful.
- $f_u \in \mathbb{R}^d$ is user $u$'s embedding, a $d$-dimensional vector encoding that user's latent viewpoint or perspective.
- $f_n \in \mathbb{R}^d$ is note $n$'s embedding, a $d$-dimensional vector encoding that note's latent viewpoint or perspective.
- $f_u \cdot f_n$ denotes the inner product $\sum_{j=1}^{d} f_u^{(j)} f_n^{(j)}$.

The parameters $\{\mu, i_u, i_n, f_u, f_n\}$ for all users and notes are learned by minimizing a regularized squared error objective:

$$\sum_{r_{un}} \left( (r_{un} - \hat{r}_{un})^2 + \lambda_i(i_u^2 + i_n^2 + \mu^2) + \lambda_f(\|f_u\|^2 + \|f_n\|^2) \right) \qquad (2)$$

where the sum ranges over all observed ratings in the dataset, and $\lambda_i, \lambda_f > 0$ are regularization parameters.

The factor dimension $d$ is set to 1 in production for practical reasons; we discuss challenges with increasing $d$ as a potential mitigation to our attack in Section 5.2.

Notes achieve "Helpful" status and display publicly only if the intercept $i_n$ exceeds a fixed threshold $\tau$, typically set to $0.40$ in practice. This threshold criterion is central to understanding the attack: the intercept $i_n$ is high precisely when the note receives positive ratings from users spanning diverse factor positions $f_u$. A note that only appeals to a narrow coalition, even if that coalition is large and rates it unanimously helpful, will have a low intercept because the alignment term $f_u \cdot f_n$ absorbs that coalition's support. Conversely, a note with high $i_n$ must have support that cannot be explained by any single direction in factor space, satisfying the diverse agreement requirement of the system.

## 3. Synthetic Consensus Manipulation

The following demonstrates how a coordinated group could generate synthetic consensus in matrix factorization.

**Threat Model.** Consider an adversary who controls $k$ accounts on the platform and seeks to make a target note $n^*$ achieve Helpful status (i.e., $i_{n^*} > \tau$). The adversary can:

- *Create and control multiple user accounts.* Section 5.1 analytically and empirically analyzes the minimum

number of accounts needed to move the note from $i_{n^*} < \tau$ to $i_{n^*} > \tau$.

- *Observe all public information, which includes all past notes and ratings on the platform.* While the adversary does not have direct access to the learned model parameters $\{f_u, f_n, i_u, i_n\}$, they could observe or reconstruct them. For transparency, X Community Notes publishes both the code and the dataset of ratings with a 48 hour delay, allowing anyone to run the matrix factorization and estimate the parameter values.[9]

- *Submit ratings on any notes, including coordinating ratings across their controlled accounts.*[10] Community Notes' design allows users to freely select which notes they rate, rather than being limited to assigned notes.

**User Factors Are Determined by Rating History.** The synthetic consensus attack leverages a property of the matrix factorization algorithm: since user embeddings are determined by rating history, constructing the rating history for a user effectively dictates the user's position in factor space. To create synthetic consensus, the adversary needs $k$ controlled accounts to simulate positions in factor space representing differing views. The controlled accounts coordinate to rate the target note $n^*$ as helpful, creating the appearance of agreement and influencing $i_{n^*}$ above the threshold $\tau$.

**Attack Strategy.** Formally, the synthetic consensus attack proceeds in two phases:

**Phase 1: Establishing Diverse Factor Positions.** The adversary rates existing notes on the platform with the goal of positioning $k$ accounts at diverse locations in the latent factor space. Specifically, the adversary aims to establish accounts with factor vectors $\{f_{a_1}, f_{a_2}, \ldots, f_{a_k}\}$ that span the space. This phase requires the adversary to solve an inverse problem: given a desired target factor $f^{target}$ for account $a$, determine how to rate existing notes such that the matrix factorization will infer $f_a \approx f^{target}$.

**Phase 2: Coordinating Support for Target Note.** Once the controlled accounts occupy diverse factor positions, the adversary has all $k$ accounts rate the target note $n^*$ as helpful.[11] Because the algorithm has learned that these accounts represent diverse perspectives (based on their Phase 1 voting patterns), their unanimous support for $n^*$ will be interpreted as a sign of bridging agreement, and the intercept $i_{n^*}$ will

---

[9]X provides X Community Notes code and data for research purposes. Usage related to platform manipulation is a violation of X rules and developer terms.

[10]Coordinated platform manipulation is a violation of X rules and may result in accounts being suspended https://help.x.com/en/rules-and-policies/x-rules

[11]In some rare cases, it might be optimal to rate the target note as "not helpful" to increase the intercept $i_{n^*}$, see Section 5.1

be driven up. This phase does not require any technical sophistication, but rather just coordination.

The Phase 1 inverse problem can be addressed by developing a voting strategy given oracle access to note parameters[12] then predicting parameters from note text (see Section 4.2).

**Predicting Note Parameters Yields a Voting Strategy.**
Recall that the matrix factorization learns parameters by minimizing the squared prediction error over observed ratings. Suppose the adversary has oracle access to the underlying note parameters $f_n$ and $i_n$ of the note. In addition, the global intercept $\mu$ is fairly stable and can be treated as a constant. For a controlled account $a$ that the adversary wishes to position at target factor $f_a^{target}$ with target intercept $i_a^{target}$, the optimal rating behavior is straightforward: when deciding whether to rate an existing note $n$ as "helpful" (1), "somewhat helpful" (0.5), or "not helpful" (0), the account should choose the rating closest to the predicted rating

$$\hat{r}_{an} = \mu + i_a^{target} + i_n + f_a^{target} \cdot f_n \qquad (3)$$

Therefore, the key challenge is predicting $f_n$ and $i_n$ based on note content. This is empirically demonstrated in Section 4.2. Given these predictions, the adversary can compute $\hat{r}_{an}$ for any note $n$ and any desired target position $(f_a^{target}, i_a^{target})$, then vote accordingly. By systematically applying this strategy across existing notes, the adversary's controlled accounts will converge to their desired factor positions during Phase 1, enabling them to span the latent space and simulate diverse perspectives in Phase 2.

## 4. Empirical Results

An attacker can obtain a diverse range of user factors, as shown in the following simulation.

### 4.1. Experimental Setup

**Dataset.** Simulations use the publicly released X Community Notes dataset[13] as of Jan 29, 2025, which contains all notes and ratings data on X from inception (Jan 2021) to two days before the dataset release (Jan 2025).

Table 1 summarizes the distribution of ratings and statuses on the platform. Figure 2 visualizes the distributions of user and note factors and intercepts.

**Computational Resources.** All experiments were run on a single machine with 4× Intel Xeon E7-4870 CPUs (40 cores, 2.4GHz) and 1TB RAM. The total wall-clock time

*Table 1.* Distribution of ratings and note statuses on the platform. While the majority of ratings are Helpful, only a select set of notes achieve bridging helpfulness.

|  | Count | % |
|---|---|---|
| **Total Ratings** | 126.9M | 100% |
| Helpful | 75.7M | 59.7% |
| Not Helpful | 47.2M | 37.2% |
| Somewhat Helpful | 4.0M | 3.1% |

|  | Count | % |
|---|---|---|
| **Total Notes** | 1.85M | 100% |
| Currently Rated Helpful | 145K | 7.9% |
| Needs More Ratings | 1.63M | 88.1% |
| Currently Rated Not Helpful | 73.3K | 4.0% |

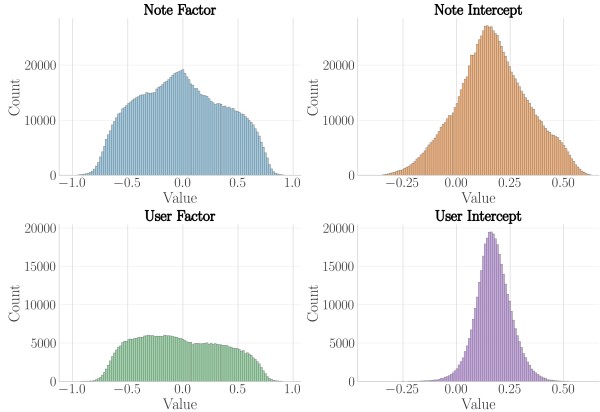

*Figure 2.* Distribution of factor values and intercepts values for all users and notes on the platform.

required for a full run of all experiments (excluding API calls) is at most 100 hours.

### 4.2. Predicting Note Parameters from Text

A critical component of the attack is the adversary's ability to predict the latent parameters $f_n$ and $i_n$ that the matrix factorization algorithm will assign to a note based solely on its text content. If these parameters can be accurately estimated before submitting the note to the system, adversaries can strategically position their controlled accounts in Phase 1 to maximally boost the target note's intercept in Phase 2.

**Model Architecture.** Given a note's text, obtain a dense embedding using the Voyage embedding model.[14] This embedding serves as input to a shallow Multilayer Perceptron (MLP) with ReLU activations that outputs predictions $\hat{f}_n \in \mathbb{R}$ and $\hat{i}_n \in \mathbb{R}$ for the note's factor and intercept,

---

[12]Since X Community Notes publishes data with a 48 hour lag, the adversary cannot observe ratings and compute parameters in real-time. Thus, the voting strategy must forecast these parameters.

[13]https://x.com/i/communitynotes/download-data

[14]Specifically, voyage-3-large model, obtaining 1024-dimensional embeddings. See https://docs.voyageai.com/docs/embeddings.

respectively.[15] The model uses only the note text itself: it ignores the content of the post being annotated and does not follow or parse any URLs referenced in the note.

**Training.** The note factor prediction model is trained on note text with gold truth[16] values for $f_n$ and $i_n$ obtained from running the open source code for the production X Community Notes system. See Appendix F for complete hyperparameter and implementation details.

**Results.** Figure 3 shows the distribution of prediction errors (actual minus predicted values) for both note intercepts $i_n$ and factors $f_n$. The residuals are sufficiently well concentrated around 0, indicating that the model achieves reasonable prediction accuracy, thereby enabling adversaries to estimate where notes will be positioned in latent space.

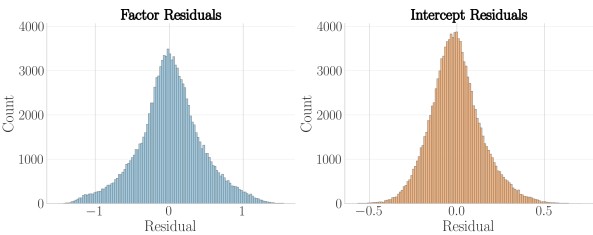

*Figure 3.* Prediction residuals for note factors and intercepts.

### 4.3. Achieving Diverse Factor Positions

Given that note parameters can be predicted with moderate accuracy, an adversary can leverage these predictions to achieve account factors throughout the factor space, validating the feasibility of Phase 1 of the manipulation.

**Experimental Setup.** In simulation, 100 adversarial accounts attempt to position themselves uniformly across the factor spectrum. Each adversary $a_i$ is assigned a target factor value $f_i^{target}$ evenly spaced in $[-0.5, 0.5]$ (covering the majority of the range of observed user factors in the real data). As per the voting strategy described in Section 3, each adversary rates a randomly sampled set of 100 existing notes. For each note $n$, the adversary chooses the rating $r_{an}$ (0, 0.5, or 1) that minimizes the error $|r_{an} - (\mu + i_a^{target} + \hat{i}_n + f_a^{target} \cdot \hat{f}_n)|$, where $\hat{i}_n$ and $\hat{f}_n$ are the predicted parameters from the model in Section 4.2. $i_a^{target}$ is set to the population mean intercept value for all adversaries. After these strategic votes are added to

the dataset, matrix factorization produces the learned factors $\{f_{a_1}, \ldots, f_{a_{100}}\}$ for the adversarial accounts.

[15]External training of models on note text is in violation of X developer terms https://docs.x.com/developer-terms/agreement

[16]Following crowdsourcing terminology, we say "gold truth" rather than "ground truth" because these values are not objective facts about, say, the note quality, but outputs of the production scoring algorithm itself—which is precisely what an attacker needs to predict.

**Results.** Figure 4 shows the distribution of achieved factor values for all 100 adversarial accounts after applying the strategic voting protocol. The histogram demonstrates that adversaries obtain factors spanning $[-0.4, 0.4]$.

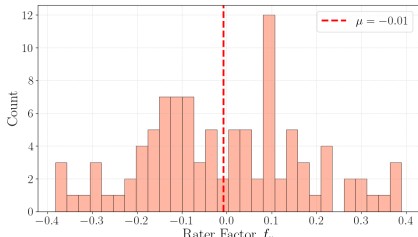

*Figure 4.* Distribution of achieved factor values for 100 adversarial users after voting strategically.

## 5. Quantifying Attack Effort

Given that adversaries can position themselves across the factor spectrum, Phase 2 estimates the effort required to manipulate note statuses as the *Manipulation Resistance Score* (MRS): the minimum number of additional ratings required to drive a note $n$ with current intercept $i_n$ above the helpfulness threshold $\tau$.[17] Empirically, MRS is shown to be $<10$ for up to 10.7% of notes with intercepts below the median. The analysis on MRS also uncovers a counterintuitive property of the algorithm: in certain cases, rating a note as "Not Helpful" can sometimes *increase* its intercept. The MRS metric is incorporated into a broader cost model to characterize the feasibility of the attack and identify potential mitigations.

### 5.1. Manipulation Resistance Score

Computing the exact MRS requires solving a combinatorial optimization problem over all possible sets of injected ratings. The MRS can instead be approximated via a greedy algorithm: at each iteration, inject the one rating that maximally increases the note intercept; repeat until $i_n > \tau$. The number of iterations required is the estimate of the MRS.

**Optimal Single Rating Injection.** The core subroutine is determining, given a note's current set of ratings, the optimal rating $(f_u^*, r^*)$ to inject: that is, the user factor $f_u^*$ and rating value $r^* \in \{0, 0.5, 1\}$ that maximally increases the note intercept $i_n$.

Since the production X Community Notes system uses a single latent dimension ($d = 1$), it is possible to derive

closed-form expressions for this optimization. For a note $n$ with current parameters $(f_n, i_n)$ learned from $N$ existing ratings, adding a new rating from a user with factor $f_u$ induces a rank-one update to the Gram matrix of the underlying ridge regression. Applying the Sherman-Morrison formula, the change in note intercept takes the form:

[17]The MRS is computed against a static set of ratings and serves as a measure of susceptibility of the note to manipulation, but does not account for potential dynamics and feedback loops after the note is surfaced on the platform. See discussion in Section 6.1.

$$\Delta i_n(f_u, r) = \frac{(\alpha - \beta f_u) \cdot r + p(f_u)}{q(f_u)} \quad (4)$$

where $\alpha$ and $\beta$ depend on the second and first moments of the existing rater factor distribution respectively, and $p(f_u)$ and $q(f_u)$ are polynomials in the user factor $f_u$, with coefficients determined by summary statistics from the existing ratings and initial note parameters (see Appendix A.5 for explicit expressions).

In order to find the maximal change in intercept, first iterate over the possible rating choices $\{0, 1\}$.[18] For a particular choice of $r^*$, the optimal user factor $f_u^*$ is found by maximizing $\Delta i_n$ over a feasible range of $f_u \in [-0.4, 0.4]$ (*narrow*, representing factors demonstrated in the previous section) or $[-1, 1]$ (*wide*, representing factors present in the true distribution). Setting the derivative to zero yields a quadratic equation whose solution, clipped to the feasible region, gives the optimal injection point in closed-form. Full derivations are found in Appendix A.

**Computing MRS.** The experimental setup is as follows: Given the closed-form optimal injection, compute MRS on all notes that were less than 2 weeks old (notes older than 2 weeks have their statuses locked) as of the data date by iteratively: (1) finding the optimal rating to inject, (2) adding it to the rating set, (3) re-solving for note parameters [19], and (4) repeating until $i_n > \tau$. The global intercept constant and rater intercept are held constant at .17 (population mean).[20]

Figure 5 shows the distribution of MRS values across notes as a function of the initial note intercept and rating volume. MRS can be $<10$ for many notes, particularly with intercepts in $[0.2, 0.4]$, even despite high rating volumes (see Table 2). As expected, notes with a lower intercept or higher rating volume tend to be harder to manipulate.

The same attack would also apply to adversarially voting on a note that is currently rated "Helpful": rather than max-

---

[18]We do not consider $r = 0.5$ as a candidate since it is always suboptimal: since $\Delta i_n$ in Eq. 4 is an affine function of $r$ (holding all else constant), it is always optimized at an extreme point (0 or 1 in our case).

[19]All MRS values use X's production regularization convention—penalties scaled by the dataset-wide mean rating count rather than each note's; see Section B.

[20]The strategy is additionally stress tested against the imprecision of factor generation in the previous section by choosing a random point in the distribution within .15 of the target factor. In 99% of cases this results in 1 or 0 additional ratings required relative to the greedy optimal strategy.

*Table 2.* Percentage of notes by initial intercept that can be manipulated with $<10$ additional ratings, when rater factors are selected from $\in [-0.4, 0.4]$ or $\in [-1, 1]$

| Percent of notes with MRS $<10$ | [-0.4, 0.4] | [-1, 1] |
|---|---|---|
| Intercept $<$-0.05 (Not Helpful Threshold) | 0.0% | 0.0% |
| Intercept $<$0.13 (25th Percentile) | 0.5% | 1.0% |
| Intercept $<$0.24 (50th Percentile) | 7.0% | 10.7% |
| Intercept $<$0.4 (Helpful Threshold) | 27.2% | 31.6% |

imizing $\Delta i_n$ (Eq. 4), the adversary minimizes it. See Appendix C for the corresponding analysis.

**An Analytical Insight.** Equation 5 also reveals a counter-intuitive insight: *it is theoretically possible that the optimal rating to increase a note intercept is "Not Helpful"*, depending on the geometry of the existing ratings.

The key observation is that in Equation 4 the rating value $r \in \{0, 1\}$ enters only through the numerator term $(\alpha - \beta f_u) \cdot r$. Therefore, to maximize $\Delta i_n$, the optimal rating is determined entirely by the sign of $(\alpha - \beta f_u)$:

$$r^* = \begin{cases} 1 \text{ (Helpful)} & \text{if } \alpha - \beta f_u \geq 0 \\ 0 \text{ (Not Helpful)} & \text{if } \alpha - \beta f_u < 0 \end{cases} \quad (5)$$

When existing raters are tightly clustered on one side of the factor spectrum, an extreme voter from the *same* side rating a note as "Not Helpful" can drive up the note intercept. When holding user parameters constant, computing note parameters amounts to fitting a line to the (user factor, rating) pairs, where the slope is the note factor $f_n$ and the $y$-intercept is $i_n$. When users clustered around e.g. $f_u = -0.5$ all give high ratings, adding a single user at $f_u = -1$ who rates artificially low steepens the slope of the best-fit line, raising the $y$-intercept. See Appendix A.6 for an explicit characterization and concrete numerical example. This is arguably an undesirable property: rating a note as "Not Helpful" should not drive up the helpfulness score.

### 5.2. Cost Model for the Full Attack

Given MRS as a measure of manipulation difficulty, we model the cost for manipulating a *single* note as:

$$C = c_m + n_v \cdot n_a \cdot c_a + n_v \cdot c_v \quad (6)$$

where $c_m$ is the one-time cost to train a prediction model, $c_a$ is the cost to create and maintain an eligible account, $c_v$ is the cost to cast a single rating, $n_v$ is the number of ratings required to manipulate a note (i.e., the MRS), and $n_a$ is the number of distinct accounts required per vote. This cost model is intentionally a simple, interpretable abstraction of the key levers that influence manipulation difficulty, rather than a fully general attacker utility model. The model's purpose is to identify which components dominate cost

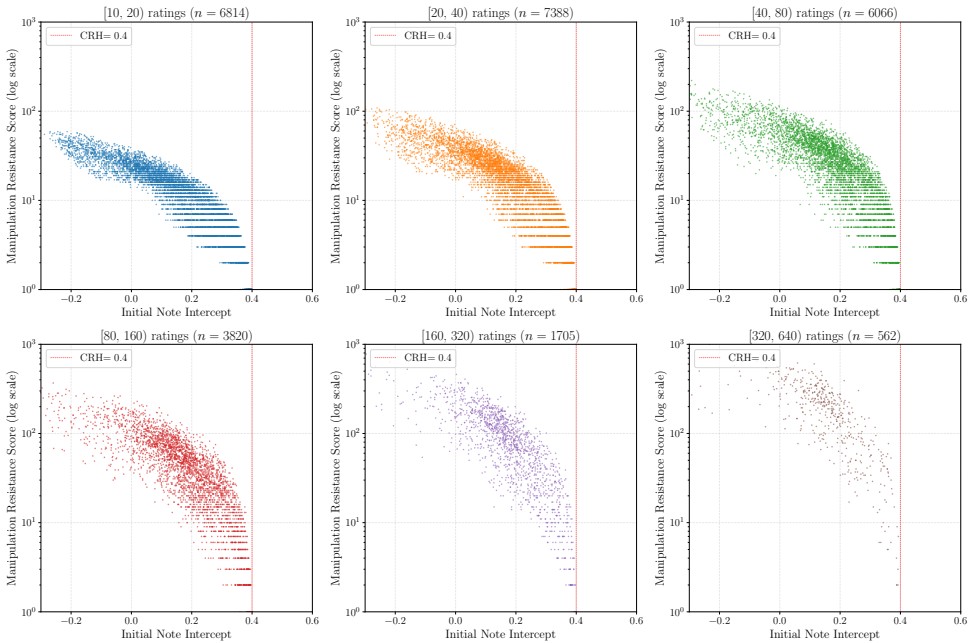

*Figure 5.* Distribution of MRS across notes as a function of initial rating volume and intercept of the note for rater factors $\in [-0.4, 0.4]$

and enable comparison of mitigations. The model rests on three assumptions—additive decomposition across attack phases, linearity in per-unit costs, and anonymity of attacker accounts—each formally stated and justified in Appendix E. A discussion of each component and mitigations follows.

**Model Training Cost ($c_m$).** Because X Community Notes publishes its source code and rating data, adversaries can train models to estimate note parameters before voting. This cost is negligible: embedding all notes costs at most $20 using commercial APIs, though providers like Voyage AI offer 200 million tokens free, which is more than sufficient for the entire dataset. Training the lightweight MLP on top requires only a few CPU/GPU-hours, which is negligible.

*Mitigation:* This cost cannot be meaningfully increased without reducing X Community Notes' commitment to transparency and open data. In fact, $c_m$ may even *decrease* over time as models become more capable and efficient.

**Account Maintenance Cost ($c_a$).** X Community Notes requires accounts to verify a phone number from a trusted carrier to be eligible for rating.[21] Based on low-cost prepaid mobile plans, $c_a \approx \$3$/month (see Appendix D).

*Mitigation:* Strengthening identity verification requirements (e.g., requiring older accounts or stricter carrier verification) directly increases $c_a$. However, this creates tension with

accessibility goals by excluding legitimate contributors.

**Vote Casting Cost ($c_v$).** The cost per vote is minimal: for example, at 10 seconds per rating and California minimum wage, $c_v \approx \$0.05$. Automation could reduce this to near-zero, though no Community Notes product currently offers a public API for submitting ratings.

*Mitigation:* Adding CAPTCHAs or other anti-automation measures increases $c_v$, but introduces friction that may reduce legitimate participation and slow ratings activity. Such interventions are counterproductive as they conflict with the goal of rapidly addressing misinformation.

**Number of Votes Required ($n_v$).** This corresponds to the MRS analyzed in Section 5.1. Empirically, $n_v$ varies depending on the current rating distribution.

*Mitigation:* Increasing the helpfulness threshold $\tau$, or requiring more ratings before a note can achieve Helpful status can increase $n_v$. However, such mitigations are at odds with showing as many high quality notes as quickly as possible. Another potential mitigation is to increase the dimensionality of the factor space. However, higher-dimensional embeddings introduce their own challenges: the algorithm may learn to bridge across dimensions that do not correspond to meaningful ideological divisions, such as writing style, tone, or use of humor. In practice, $d = 1$ has proven effective because it captures the dominant axis of disagreement, typically corresponding to political orientation.

---

[21]In addition to other requirements, see https://communitynotes.x.com/guide/en/contributing/signing-up

**Accounts Per Vote** ($n_a$). Currently, any account can cast one vote on any note, so $n_a = 1$.

*Mitigation:* Rather than allowing any account to vote on any note, the system could restrict voting on each note to a random subset of eligible users. To avoid reducing overall rating volume, a practical implementation could elicit ratings from a random sample of users for each note while continuing to allow all users to rate freely, but stop notes from being marked as helpful when the intercept computed from the population-sampled subset significantly deviates from the intercept computed on all ratings. This is a promising mitigation since it increases attack cost linearly with population size without reducing system accessibility or slowing note evaluation. We deployed this form of mitigation in X's Community Notes system.

**Cost Estimate.** Consider an adversary seeking to manipulate the bridging score for a single note. As previously established, given constraints on the visibility and original note intercept, this can require less than 10 ratings. The cost of this potential manipulation using representative estimates of costs for existing systems with $n_a = 1$, $c_a = \$3/month$, $c_v \approx \$0.05$, and $c_m \approx \$0$ is estimated:

$$C \approx \underbrace{0}_{c_m} + \underbrace{10 \times 1 \times 3}_{\text{accounts}} + \underbrace{10 \times 0.05}_{\text{votes}}$$
$$= \$0 + \$30 + \$0.50 = \$30.50.$$

The **dominant cost is account maintenance**, while both model training (which is a one-time cost) and vote casting are negligible. This also illustrates why population-sampled intercepts are particularly effective: a 1% sampling rate would require $100\times$ as many accounts, increasing the per note manipulation cost hundred-fold.

In summary, this analysis reveals a tension in defending Community Notes: certain mitigations that increase attack cost may also reduce the speed at which the system surfaces helpful context on potentially misleading posts.

## 6. Discussion

### 6.1. Limitations

Our analysis is subject to several limitations that should be considered when interpreting the results, as well as a mitigation deployed at X as described below.

**Anti-abuse algorithm components** Although voting cost is negligible and adversarial accounts could theoretically be reused to manipulate multiple notes, thereby amortizing the account maintenance cost, production implementations of Community Notes can include practical guardrails on top of the core matrix factorization that impact the cost of the attack, particularly at scale. For example, the abuse safeguards for X Community Notes include:

- **Correlated Rater Detection** with several approaches to identify anomalous engagement. The effects can include dropping ratings entirely or re-computing the matrix factorization without anomalous ratings to identify notes with large intercept changes.

- **Rater Engagement Intercept** re-computes note intercepts, excluding the 0.1% most active raters. Notes scoring below a safeguard must gather more ratings.

- **Net Helpful Minimums** require that notes shown as Helpful have at least 4 (sometimes greater) Helpful ratings from both positive and negative factor raters.

The three components above interact with the two-phase attack of Section 3 in different ways. Correlated Rater Detection examines co-rating patterns across users and is the safeguard most directly aligned with the attack's signature: as the same coalition of accounts is reused across multiple target notes, the resulting co-rating structure becomes increasingly visible. In addition, Rater Engagement Intercept becomes effective at scale as reused accounts accumulate into the excluded 0.1% fraction of most active raters. Evading these filters over longer time horizons requires more sophistication in the rating patterns exhibited by attackers, thereby potentially increasing the effective number of accounts required and increasing the cost of sustained attacks at scale. Net Helpful Minimums require Helpful ratings from raters of both factor signs, which Phase 1 already produces by design, so this safeguard adds at most a small additive constant to $n_v$. Beyond these components, X Community Notes also includes other mechanisms that use *rating tags* which are not incorporated into the matrix factorization to identify notes as inaccurate, abusive, etc.

**Population sample filtering** In addition to the existing components above, we deployed an additional mitigation on X Community Notes: population sample filtering. People often rate notes when they encounter them in normal usage of the app. However, contributors on X also rate notes *in response to* notifications or a "Needs Your Help" feed. Since the matrix factorization attack presented in this work depends on attackers being able to rate targeted notes, population sample filtering recomputes quality scores on the subset of ratings *solicited from contributors*. Quality score deltas exceeding a safeguard threshold block notes from Helpful display status, thereby increasing the effective cost of the attack as modeled by Section 5.2.

**Feedback Dynamics** Beyond explicit safeguards, this work examines note status in a static setting, where the

attacker is the only party that introduces additional ratings. The prominence and platform visibility associated with Helpful note status attracts additional ratings from contributors, which can result in a low quality note losing Helpful status as the Community Notes algorithm runs with the added contributor ratings. In effect, operating in a dynamic environment increases the potential cost to the attacker, who must counteract any additional ratings from Community Notes contributors.

**Conservative Approximations in Computing MRS** On the other hand, this analysis makes some conservative approximations in computing the cost of fabricating synthetic consensus. First, the minimum number of ratings required may be lower when computing MRS via exact combinatorial optimization problem rather than a greedy algorithm. Second, adversaries optimizing user intercepts jointly with factor positions rather than fixing them at the population mean would gain additional leverage over the note intercept (for instance, by cultivating a history of harsh ratings to achieve a low $i_u$). Third, the note parameter prediction model uses only the note text itself, ignoring the content of the annotated post and any referenced URLs. Incorporating this additional context could improve prediction accuracy and further reduce the number of ratings needed in Phase 1. Similarly, strategically selecting which notes to rate (rather than voting on an arbitrary subset) could accelerate factor positioning and reduce the overall attack cost.

### 6.2. Additional Related Work

Prior work on manipulation of crowdsourced fact-checking has focused on earlier, non-bridging systems. Mujumdar & Kumar (2021) studied adversarial attacks on an early version of Birdwatch (the initial name of X's Community Notes) that used a simple majority threshold rather than matrix factorization; their findings do not transfer to the bridging-based algorithm we analyze here. In concurrent work, Truong et al. (2025) use counterfactual rater behavior simulations to quantify error rates in publishing helpful versus unhelpful notes on Community Notes. Our work complements this work by considering a novel attack to strategically fabricate diverse agreement and potentially target individual notes.

The matrix factorization framework underlying our analysis has long been studied in the recommender systems literature, where adversarial manipulation is an established concern. As hinted in Section 2, Community Notes and matrix factorization-based recommender systems are similar primarily at the modeling level: both learn latent representations of users and items from a sparse user-item matrix. The subsequent goals of the two systems are largely different. In the recommender setting, matrix factorization is predominantly used for personalized prediction or ranking based on relative positions in latent space; Community Notes instead uses the note intercept as a single global score of note quality, departing from the traditional use of matrix factorization. Consequently, the attack objectives differ as well. Classic recommender-system shilling attacks aim to push a target item, nuke a target item, or, more generally, compromise recommendation integrity by injecting fake user profiles, including through well-known heuristic attack families such as random, average, bandwagon, and segment attacks (Lam & Riedl, 2004; Mobasher et al., 2005; 2007). In contrast, the natural adversarial objective in Community Notes is less to "promote one item" than to cultivate accounts that can fabricate diverse agreement and potentially manipulate many arbitrary notes. Differences also surface at the mechanism level: Community Notes presents an online problem where, unlike classical recommender systems, users cannot inject arbitrary rows into the matrix (e.g., rate all available items on a platform), but can merely add ratings to "new items" to manipulate them. Moreover, attacks in modern recommender settings, such as PoisonRec (Song et al., 2020), look quite different because they operate in an implicit-feedback setting rather than one with binary votes. Finally, it is worth noting that Community Notes provides an unusually white-box setting for the attacker, since both the scoring code and complete data (with a 48 hour delay) are published, whereas many popular large-scale recommender systems (e.g., Netflix, Amazon) are comparatively more black-box.

Lastly, adversarial manipulation of voting-based evaluation systems has also been studied in the context of LLM leaderboards, where strategic voting can artificially inflate model rankings (Huang et al., 2025; Zhao et al., 2025).

## 7. Conclusion

We have shown the degree to which the matrix factorization portion of Community Notes is vulnerable to coordinated manipulation: by strategically positioning accounts across the latent factor space, adversaries can attempt to fabricate synthetic consensus. Using historic production data, we find that up to 10.7% of lower quality notes could be manipulated above consensus thresholds using less than 10 ratings. To mitigate such attacks, Community Notes deployments can include additional components beyond the matrix factorization, such as the population sample filtering approach developed and deployed on X as part of this work, which can support note quality and limit the effectiveness of attacks.

More broadly, our approach leverages a fundamental property of the system – user factors are determined by rating history – which is central to how bridging-based consensus mechanisms model differences in perspectives. We call for further development of practical defenses that can mitigate manipulation, while maintaining the transparency and openness that enabled both this and other public research exploring, critiquing, and strengthening Community Notes.

## Impact Statement

This work was conducted with the goal of proactively identifying vulnerabilities in Community Notes before they can be exploited in the wild. This work was conducted as an open research collaboration between Stanford University and the X Community Notes team, leveraging the benefits of X's open source, open data approach to realize improvements to Community Notes through public contributions. As part of the collaboration X deployed mitigations in production prior to this publication and released them as part of the open-source algorithm. All experiments, including those involving real-world production data, were conducted in a simulated environment using the official open-source codebase and did not affect the live platform in any way; we did not manipulate any notes on the actual live system. We hope this work spurs further research into potential vulnerabilities and defenses, ultimately strengthening a valuable tool that is seeing increasing adoption across platforms as a scalable approach to combating misinformation.

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

# A. Optimal Rating Injection: Full Derivation

This appendix provides the complete derivation of the optimal rating injection formula presented in Section 5.1. Given a set of ratings on a note, we derive a closed-form expression for the optimal rating to inject to increase the intercept as much as possible.

## A.1. Notation

For the sake of reader understanding, we use streamlined notation for the derivation that better mimics standard regression notation and maps to the main paper as follows:

| Appendix | Main Paper | Description |
|---|---|---|
| $x_i$ | $f_{u_i}$ | Factor of user $i$ |
| $y_i$ | $r_{u_i n} - \mu - i_{u_i}$ | Adjusted rating (rating minus global and user intercepts) |
| $m$ | $f_n$ | Note factor (slope) |
| $b$ | $i_n$ | Note intercept |
| $\lambda_m$ | $\lambda_f$ | Factor regularization parameter |
| $\lambda_b$ | $\lambda_i$ | Intercept regularization parameter |
| $N$ | – | Number of ratings on the note |

Thus $(x_i, y_i)$ refers to the $i$-th rating. We assume by default that the rater has the mean intercept of $0.17$ and that $\mu = 0.17$. The adjusted rating $y$ effectively equals the rating ($1$ or $0$) minus $0.17$ (for global intercept) minus $0.17$ (for the user intercept).

The optimization objective is:

$$L(m, b) = \frac{1}{N} \sum_{i=1}^{N} (y_i - (m x_i + b))^2 + \lambda_m m^2 + \lambda_b b^2.$$

## A.2. Closed Form for the Original Optimization Problem

Consider the following useful summary statistics:

$$\Sigma_1 = \sum_{i=1}^{N} x_i, \qquad \Sigma_2 = \sum_{i=1}^{N} x_i^2.$$

The normal equation for this ridge regression problem is:

$$(\mathbf{X}^\top \mathbf{X} + N \mathbf{\Lambda}) \, \boldsymbol{\theta} \; = \; \mathbf{X}^\top \mathbf{y}$$

where:

- $\mathbf{X}$ is the $N \times 2$ matrix with $X_{i,1} = x_i$ and $X_{i,2} = 1$

- $\boldsymbol{\theta} = \begin{bmatrix} m \\ b \end{bmatrix}$ is the parameter vector (slope and intercept)

- $\mathbf{\Lambda} = \begin{bmatrix} \lambda_m & 0 \\ 0 & \lambda_b \end{bmatrix}$ is the regularization matrix

- $\mathbf{y} = \begin{bmatrix} y_1 \\ \vdots \\ y_N \end{bmatrix}$ is the vector of target values

For convenience, we write:

$$\mathbf{S} \ = \ \mathbf{X}^\top \mathbf{X} + N\mathbf{\Lambda} \ = \ \begin{bmatrix} A & B \\ B & C \end{bmatrix}$$

where:

$$A = \Sigma_2 + N\lambda_m, \quad B = \Sigma_1, \quad C = N(1 + \lambda_b),$$

with determinant $D = AC - B^2 \geq 0$.

### A.3. Change in Intercept After Injecting a Rating

Assume we append a single new rating $(x, y)$ to the existing $N$ points. Let

$$\mathbf{X}' \ = \ \begin{bmatrix} \mathbf{X} \\ x & 1 \end{bmatrix}, \qquad \mathbf{y}' \ = \ \begin{bmatrix} \mathbf{y} \\ y \end{bmatrix}.$$

**Effect on $\mathbf{X}^\top\mathbf{X}$.** Writing $\mathbf{v} := \begin{bmatrix} x \\ 1 \end{bmatrix}$ for the new row of the input matrix, the Gram matrix after augmentation is the rank-one update

$$\mathbf{X}'^\top\mathbf{X}' \ = \ \mathbf{X}^\top\mathbf{X} + \mathbf{v}\mathbf{v}^\top.$$

**Effect on the normal equation.** Because the mean-squared-error term is now averaged over $N + 1$ points, the ridge-regression normal equation becomes

$$\left(\mathbf{X}'^\top\mathbf{X}' + (N+1)\mathbf{\Lambda}\right)\boldsymbol{\theta}' \ = \ \mathbf{X}'^\top\mathbf{y}' \ = \ \mathbf{X}^\top\mathbf{y} + \mathbf{v}\,y.$$

Define

$$\mathbf{S}' \ := \ \mathbf{X}'^\top\mathbf{X}' + (N+1)\mathbf{\Lambda} \ = \ \underbrace{(\mathbf{X}^\top\mathbf{X} + N\mathbf{\Lambda}) + \mathbf{\Lambda}}_{=:\mathbf{S}_1} + \mathbf{v}\mathbf{v}^\top.$$

With the notation from the previous section:

$$\mathbf{S}_1 = \begin{bmatrix} A + \lambda_m & B \\ B & C + \lambda_b \end{bmatrix} =: \begin{bmatrix} A_1 & B \\ B & C_1 \end{bmatrix}, \tag{7}$$

with determinant $D_1 = A_1 C_1 - B^2 > 0$.

**Effect on the inverse of the Gram matrix.** The matrix $\mathbf{S}'$ is a rank-one update of $\mathbf{S}_1$, so the Sherman-Morrison formula gives

$$\mathbf{S}'^{-1} \ = \ \mathbf{S}_1^{-1} \ - \ \frac{\mathbf{S}_1^{-1}\mathbf{v}\mathbf{v}^\top\mathbf{S}_1^{-1}}{1 + \gamma}, \qquad \gamma := \mathbf{v}^\top\mathbf{S}_1^{-1}\mathbf{v}.$$

Because

$$\mathbf{S}_1^{-1} = \frac{1}{D_1}\begin{bmatrix} C_1 & -B \\ -B & A_1 \end{bmatrix},$$

we have that

$$\gamma = \mathbf{v}^\top\mathbf{S}_1^{-1}\mathbf{v} = \frac{1}{D_1}\left(C_1 x^2 - 2Bx + A_1\right).$$

**How does the intercept change after adding this new rating?** For convenience in calculation, let

$$\mathbf{w} \ := \ \mathbf{S}_1^{-1}\mathbf{v} \ = \ \frac{1}{D_1}\begin{bmatrix} C_1 x - B \\ A_1 - Bx \end{bmatrix}, \qquad \text{so that} \qquad \gamma \ = \ \mathbf{v}^\top\mathbf{w} > 0.$$

The right-hand side of the normal equation is

$$\mathbf{r} \ := \ \mathbf{X}'^\top\mathbf{y}' = \mathbf{X}^\top\mathbf{y} + y\mathbf{v} = \mathbf{S}\,\boldsymbol{\theta} + y\mathbf{v} = \left(\mathbf{S}_1 - \mathbf{\Lambda}\right)\boldsymbol{\theta} + y\mathbf{v}.$$

Because $\boldsymbol{\theta}' = \mathbf{S}'^{-1}\mathbf{r}$ and $\mathbf{S}'^{-1} = \mathbf{S}_1^{-1} - \dfrac{\mathbf{w}\mathbf{w}^\top}{1+\gamma}$, we decompose the product term by term:

$$\mathbf{T}_1 = \mathbf{S}'^{-1}\mathbf{S}_1\boldsymbol{\theta} = \boldsymbol{\theta} - \frac{\mathbf{w}(\mathbf{v}^\top\boldsymbol{\theta})}{1+\gamma}$$

$$\mathbf{T}_2 = -\mathbf{S}'^{-1}\boldsymbol{\Lambda}\boldsymbol{\theta} = -\mathbf{S}_1^{-1}\boldsymbol{\Lambda}\boldsymbol{\theta} + \frac{\delta}{1+\gamma}\mathbf{w} \quad \text{where } \delta := \mathbf{w}^\top\boldsymbol{\Lambda}\boldsymbol{\theta} = \frac{\lambda_m(C_1 x - B)m + \lambda_b(A_1 - Bx)b}{D_1}$$

$$\mathbf{T}_3 = y\,\mathbf{S}'^{-1}\mathbf{v} = \frac{y}{1+\gamma}\mathbf{w}$$

Combining $\mathbf{T}_1$, $\mathbf{T}_2$, and $\mathbf{T}_3$, we get:

$$\boldsymbol{\theta}' = \boldsymbol{\theta} - \mathbf{S}_1^{-1}\boldsymbol{\Lambda}\boldsymbol{\theta} \quad + \frac{-\mathbf{v}^\top\boldsymbol{\theta} + \delta + y}{1+\gamma}\mathbf{w}.$$

Note that $y - \mathbf{v}^\top\boldsymbol{\theta}$ is the difference between true and predicted rating of the new point based on our old parameters. Define this residual:

$$R := y - (mx + b).$$

We are only interested in the intercept which is the second element of $\boldsymbol{\theta}'$.

**Change in the intercept.** Write $\Delta b := b' - b$. From the vector update above we have

$$\Delta b = -\left(\mathbf{S}_1^{-1}\boldsymbol{\Lambda}\boldsymbol{\theta}\right)_2 + \frac{A_1 - Bx}{D_1}\frac{y - (mx + b) + \delta}{1+\gamma}.$$

We now evaluate each term. First:

$$\mathbf{S}_1^{-1}\boldsymbol{\Lambda}\boldsymbol{\theta} = \frac{1}{D_1}\begin{bmatrix} C_1 & -B \\ -B & A_1 \end{bmatrix}\begin{bmatrix} \lambda_m m \\ \lambda_b b \end{bmatrix} = \frac{1}{D_1}\begin{bmatrix} C_1\lambda_m m - B\lambda_b b \\ -B\lambda_m m + A_1\lambda_b b \end{bmatrix}.$$

Hence the second element is

$$\left(\mathbf{S}_1^{-1}\boldsymbol{\Lambda}\boldsymbol{\theta}\right)_2 = \frac{-B\lambda_m m + A_1\lambda_b b}{D_1}.$$

Recall that $\gamma = \dfrac{C_1 x^2 - 2Bx + A_1}{D_1}$. Therefore

$$1 + \gamma = \frac{D_1 + A_1 + C_1 x^2 - 2Bx}{D_1}.$$

Putting everything together gives

$$\Delta b = \underbrace{\frac{B\lambda_m m - A_1\lambda_b b}{D_1}}_{(\mathrm{I})} + \underbrace{\frac{A_1 - Bx}{D_1}\frac{R + \dfrac{\lambda_m m(C_1 x - B) + \lambda_b b(A_1 - Bx)}{D_1}}{\dfrac{D_1 + A_1 + C_1 x^2 - 2Bx}{D_1}}}_{(\mathrm{II})}.$$

Simplify (II):

$$(\mathrm{II}) = \frac{A_1 - Bx}{D_1}\frac{D_1 R + \lambda_m m(C_1 x - B) + \lambda_b b(A_1 - Bx)}{D_1 + A_1 + C_1 x^2 - 2Bx}.$$

Combining (I) and (II) with the common denominator $M := D_1(D_1 + A_1 + C_1x^2 - 2Bx)$:

$$(I) = \frac{(B\lambda_m m - A_1\lambda_b b)(D_1 + A_1 + C_1x^2 - 2Bx)}{M},$$

$$(II) = \frac{(A_1 - Bx)(D_1 R + \lambda_m m(C_1 x - B) + \lambda_b b(A_1 - Bx))}{M}.$$

Hence $\Delta b = \frac{N_1 + N_2}{M}$ where

$$N_1 = (B\lambda_m m - A_1\lambda_b b)(D_1 + A_1 + C_1x^2 - 2Bx),$$

$$N_2 = (A_1 - Bx)\Big[D_1 R + \lambda_m m(C_1 x - B) + \lambda_b b(A_1 - Bx)\Big].$$

Grouping terms, the $\lambda_m m$ coefficient is:

$$C_{\lambda_m} := B(D_1 + A_1 + C_1x^2 - 2Bx) + (A_1 - Bx)(C_1 x - B)$$
$$= BD_1 - B^2 x + A_1 C_1 x = D_1(B + x).$$

The $\lambda_b b$ coefficient is:

$$C_{\lambda_b} := -A_1(D_1 + A_1 + C_1x^2 - 2Bx) + (A_1 - Bx)^2 = -D_1(A_1 + x^2).$$

Putting it all together:

$$\Delta b = \frac{N_1 + N_2}{M} = \frac{(A_1 - Bx)R + \lambda_m m(B + x) - \lambda_b b(A_1 + x^2)}{D_1 + A_1 + C_1x^2 - 2Bx}.$$

Therefore, we finally obtain:

$$\Delta b(x, y) = \frac{(A_1 - Bx)R + \lambda_m m(B + x) - \lambda_b b(A_1 + x^2)}{D_1 + A_1 + C_1x^2 - 2Bx} \tag{8}$$

## A.4. Optimal Rating to Inject

Looking into this expression for $\Delta b$ gives us some intuition about which rating sign to use (helpful vs. not helpful). Specifically, the only term that depends on the rating sign is $(A_1 - Bx)R = (A_1 - Bx)(y - mx - b)$. Since the dependence is linear, we always want to inject a helpful rating if $A_1 - Bx \geq 0$ and a not helpful rating if $A_1 - Bx < 0$.

To find the optimal rating to inject, we explicitly consider two cases:

- Set $y = 1 - 0.17 - 0.17 = 0.66 = y_{\max}$ and optimize $\Delta b$ over $x \in [-1, 1]$ such that $A_1 - Bx \geq 0$.

- Set $y = 0 - 0.17 - 0.17 = -0.34 = y_{\min}$ and optimize $\Delta b$ over $x \in [-1, 1]$ such that $A_1 - Bx < 0$.

A.4.1. CASE 1: OPTIMIZING $x$ WITH $y = y_{\max}$

Set $y = y_{\max}$ and keep the constraint $A_1 - Bx \geq 0$. Denote

$$\varepsilon := y_{\max} - b, \qquad R(x) = \varepsilon - mx.$$

With this substitution the numerator of $\Delta b$ becomes

$$N(x) = (A_1 - Bx)R(x) + \lambda_m m(B + x) - \lambda_b b(A_1 + x^2),$$

and the denominator is

$$D(x) = D_1 + A_1 + C_1x^2 - 2Bx.$$

We want to find critical points of the function $f(x) = N(x)/D(x)$ on $[-1, 1]$. Using the quotient rule,

$$f'(x) = 0 \quad \Longleftrightarrow \quad N'(x)D(x) - N(x)D'(x) = 0.$$

Computing derivatives:

$$\begin{aligned}
N'(x) &= -BR(x) - m(A_1 - Bx) + \lambda_m m - 2\lambda_b bx \\
&= (2Bm - 2\lambda_b b)x + (-B\varepsilon - mA_1 + \lambda_m m), \\
D'(x) &= 2C_1 x - 2B.
\end{aligned}$$

Substituting $N'$, $N$, $D'$ and $D$, the cubic terms cancel, leaving the quadratic:

$$\alpha_2 x^2 + \alpha_1 x + \alpha_0 = 0,$$

with

$$\begin{aligned}
\alpha_2 &= -2B(Bm - \lambda_b b) - C_1(-A_1 m - B\varepsilon + \lambda_m m), \\
\alpha_1 &= 2\Big[(Bm - \lambda_b b)(D_1 + A_1) - C_1(A_1\varepsilon + \lambda_m mB - \lambda_b bA_1)\Big], \\
\alpha_0 &= (-A_1 m - B\varepsilon + \lambda_m m)(D_1 + A_1) + 2B(A_1\varepsilon + \lambda_m mB - \lambda_b bA_1).
\end{aligned}$$

The solution to this quadratic, after clipping to the desired constraints, gives the optimal $x$.

### A.4.2. CASE 2: OPTIMIZING $x$ WITH $y = y_{\min}$

Set $y = y_{\min}$ and impose the constraint $A_1 - Bx < 0$. Define

$$\varepsilon_{\min} := y_{\min} - b \quad (< 0), \qquad R_{\min}(x) = \varepsilon_{\min} - mx.$$

The derivative computation and subsequent algebra proceeds verbatim as in Case 1; every step is unchanged except that $\varepsilon$ is replaced by $\varepsilon_{\min}$. Thus, the optimal $x$ in this regime is obtained by solving the same quadratic equation, simply evaluating the coefficients with $\varepsilon = \varepsilon_{\min}$ and restricting to those $x$ for which $A_1 - Bx < 0$ and $x \in [-1, 1]$.

### A.4.3. WHICH RATING TO INJECT TO MINIMIZE THE INTERCEPT?

The algebra remains the same: find the roots of the same quadratic, except replace $\varepsilon$ with $\varepsilon_{\min}$ but keep the constraint $A_1 - Bx \geq 0$ for the first case, and replace $\varepsilon_{\min}$ with $\varepsilon$ and keep the constraint $A_1 - Bx < 0$ for the second case.

### A.5. Derivation of the simplified form in the main paper.

Translating Equation (8) to main paper notation via $(x, y, m, b, \lambda_m, \lambda_b) \to (f_u, r, f_n, i_n, \lambda_f, \lambda_i)$ and writing $\alpha := A_1 = \sum_{i=1}^{N} f_{u_i}^2 + (N+1)\lambda_f$ and $\beta := B = \sum_{i=1}^{N} f_{u_i}$, we get:

$$\begin{aligned}
\Delta i_n &= \frac{(\alpha - \beta f_u)(r - \mu - i_u - f_n f_u - i_n) + \lambda_f f_n(\beta + f_u) - \lambda_i i_n(\alpha + f_u^2)}{D_1 + \alpha + C_1 f_u^2 - 2\beta f_u} \\[2mm]
&= \frac{(\alpha - \beta f_u)\cdot r + (\alpha - \beta f_u)(-\mu - i_u - f_n f_u - i_n) + \lambda_f f_n(\beta + f_u) - \lambda_i i_n(\alpha + f_u^2)}{C_1 f_u^2 - 2\beta f_u + (D_1 + \alpha)} \\[2mm]
&= \frac{(\alpha - \beta f_u)\cdot r + \overbrace{(\alpha - \beta f_u)(-\mu - i_u - f_n f_u - i_n) + \lambda_f f_n(\beta + f_u) - \lambda_i i_n(\alpha + f_u^2)}^{p(f_u)}}{\underbrace{C_1 f_u^2 - 2\beta f_u + (D_1 + \alpha)}_{q(f_u)}} \\[2mm]
&= \frac{(\alpha - \beta f_u)\cdot r + p(f_u)}{q(f_u)}.
\end{aligned}$$

### A.6. Counterintuitive Example: Voting "Not Helpful" to Increase Intercept

We derive conditions under which voting "Not Helpful" is optimal for increasing a note's intercept, and provide a concrete numerical example.

**Deriving the variance bound.** From (8), the *only* term that depends on the injected rating value $y$ is

$$(A_1 - Bx)\, R \;=\; (A_1 - Bx)\big(y - (mx + b)\big),$$

Thus, voting "Not Helpful" (corresponding to a negative $y$) would result in an increase in the intercept precisely when the coefficient of $R = (A_1 - Bx)R$ is negative:

Intuitively, $B = \Sigma_1 = \sum_{i=1}^{N} x_i$ captures the first moment (mean) of existing raters' factors, while $A_1 = \Sigma_2 + (N+1)\lambda_m$ is the regularized second moment of the existing rater factors (since $\Sigma_2 = \sum_{i=1}^{N} x_i^2$). Writing the empirical mean and variance of existing factors, we get:

$$\bar{x} := \frac{1}{N}\sum_{i=1}^{N} x_i, \qquad s_x^2 := \frac{1}{N}\sum_{i=1}^{N}(x_i - \bar{x})^2,$$

so that $\Sigma_1 = N\bar{x}$ and $\Sigma_2 = N(s_x^2 + \bar{x}^2)$. Substituting, we get that

$$\begin{aligned}
A_1 - Bx &= \Sigma_2 + (N+1)\lambda_m - \Sigma_1 x \\
&= N\big(s_x^2 + \bar{x}^2 - \bar{x}x\big) + (N+1)\lambda_m \\
&= N\big(s_x^2 - \bar{x}(x - \bar{x})\big) + (N+1)\lambda_m.
\end{aligned}$$

Therefore, a necessary and sufficient condition for "Not Helpful" rating to increase the intercept is

$$s_x^2 \;<\; \bar{x}(x - \bar{x}) \;-\; \left(1 + \frac{1}{N}\right)\lambda_m. \tag{9}$$

This has a clear interpretation. The right-hand side is positive only when the injected factor $x$ lies *further* in the direction of the existing mean (same sign as $\bar{x}$ and $|x| > |\bar{x}|$), and even then the existing factors must be *sufficiently tightly clustered* (small $s_x^2$) to remain lower than the right-hand side.

**Numerical example.** For the sake of intuition, consider the case where the existing rater factors (who all rate the note as "Helpful") are most centered around $-0.5$, so that $\bar{x} = -0.5$. If we now try to inject a new rater at $x = -1$, the right-hand side of (9) becomes:

$$\bar{x}(x - \bar{x}) - \left(1 + \frac{1}{N}\right)\lambda_m = \left(-\tfrac{1}{2}\right)\left(-1 + \tfrac{1}{2}\right) - \left(1 + \frac{1}{N}\right)\lambda_m = \tfrac{1}{4} - \left(1 + \frac{1}{N}\right)\lambda_m \approx 0.25.$$

Thus, if the raters are sufficiently tightly clustered (i.e. $s_x^2 < 0.25$), the condition in (9) holds and a "Not Helpful" vote from our injected rater will increase the fitted intercept.

**Geometric intuition.** Figure 6 illustrates the geometry using an extreme example. Holding all other model parameters fixed, recovering $(f_n, i_n)$ amounts to fitting a line to the (user factor, adjusted rating) pairs, where the slope is the note factor $f_n$ and the $y$-intercept is $i_n$. When the existing raters' factors are tightly clustered (here near $f_u = -0.5$) and all rate the note as "Helpful," the best-fit line is nearly flat with a moderate intercept (dashed). Adding a single "Not Helpful" rating from an even more extreme rater at $f_u = -1$ pivots the line into a steep positive slope (solid), which actually *raises* the $y$-intercept.

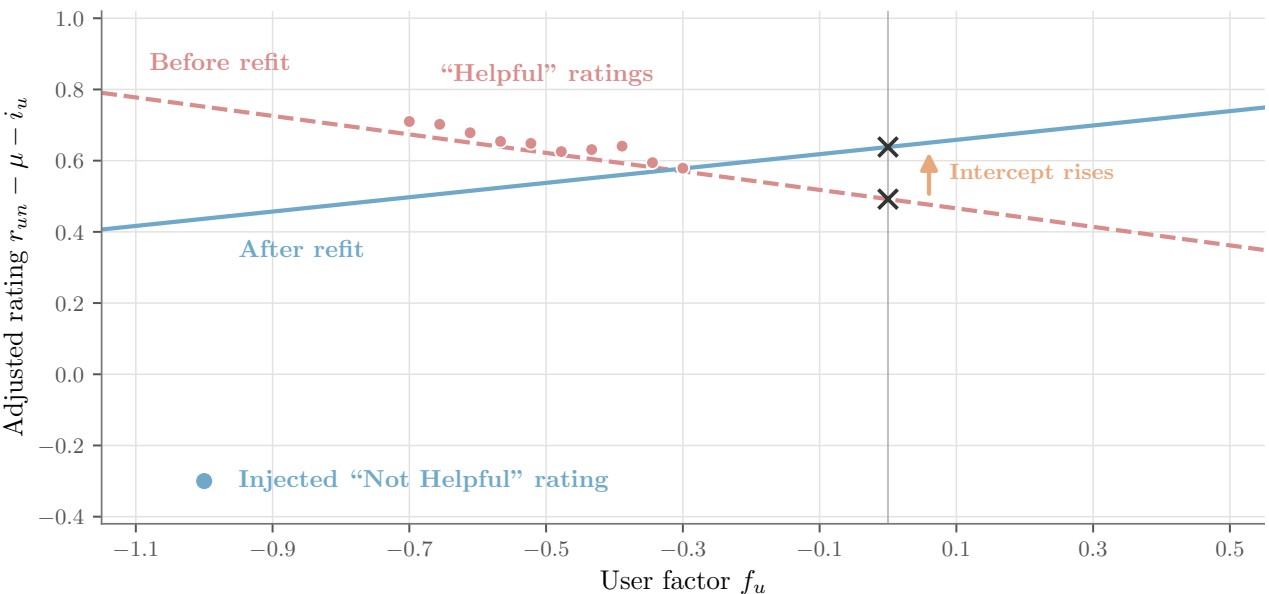

*Figure 6.* Geometric intuition for why a "Not Helpful" rating can *raise* a note's intercept. Adding a single "Not Helpful" rating from an extreme rater at $f_u = -1$ pivots the best-fit line (**dashed → solid**) and raises the $y$-intercept $i_n$.

## B. Regularization Convention for MRS Computation

The derivation in Appendix A fits, for each note,

$$L(m, b) = \sum_{i=1}^{N} (y_i - mx_i - b)^2 + K\lambda_m\, m^2 + K\lambda_b\, b^2,$$

where $K$ is a penalty multiplier. Two natural choices exist:

- *Per-note*: $K = N$, the note's own rating count.

- *Production*: $K = \bar{N}$, the dataset-wide mean ratings per note. X's production Community Notes scorer uses this convention, so the penalty is independent of $N$.

Relative to X's production convention, the per-note convention would underestimate MRS for notes with $N < \bar{N}$ and overestimate it for notes with $N > \bar{N}$. We report all MRS values under the production convention since it better captures manipulation resistance in practice.

## C. Manipulating Notes to Lose Helpful Status

The same attack would also apply to adversarially voting on a note that is currently rated "Helpful" to "take it down" and make it lose its helpful status. In terms of the optimal rating to inject, rather than maximizing $\Delta i_n$ (Eq. 4), the adversary minimizes it; this remains a second-degree expression in $f_u$ that can be easily optimized.

Figure 7 shows the corresponding distribution of MRS values, where we aim to lower the intercepts of notes with intercepts above $\tau$. Once again, we observe that many notes can be brought down to below the threshold $\tau$ with a handful of adversarial votes.

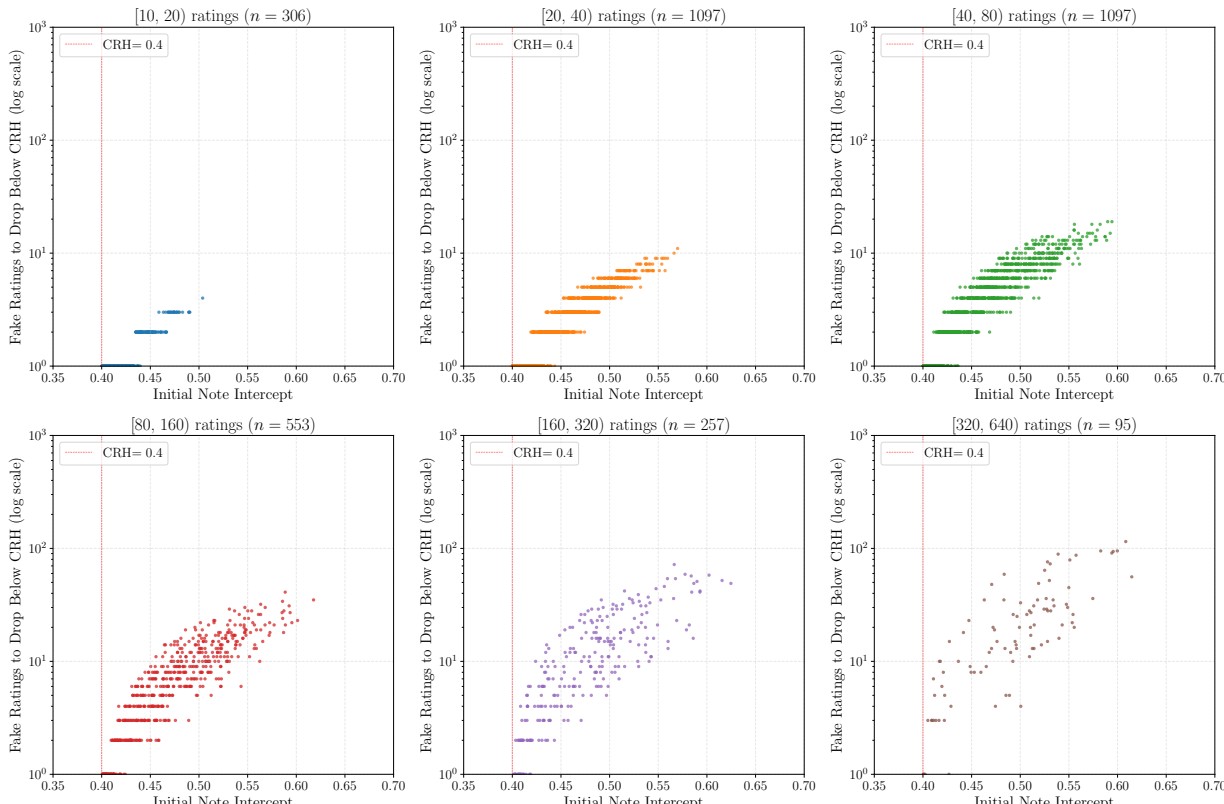

*Figure 7.* Distribution of MRS when the adversary aims to take down notes: minimum number of adversarial ratings required to drive a note's intercept from above $\tau$ to below the threshold.

It is worth emphasizing, however, that we need to be careful when interpreting these results, since we only study the attack on the core matrix factorization component. The production implementations of these matrix factorization systems often have additional safeguards and features that make a direct extrapolation from our study to the exact number of votes required tricky. For instance, X's implementation of Community Notes has a feature called *CRH inertia*, that requires that "Helpful" notes drop below a note intercept threshold that is lower than the original threshold $\tau$ in order to lose their helpfulness status and stop being shown broadly. For instance, in some cases, a note that gained "Helpful" status by reaching a note intercept of $\tau = 0.40$ must drop below $\tau' = 0.39$ in order to lose its helpfulness status. Such a guardrail might lead us to underestimate MRS.

On the other hand, X's implementation of Community Notes also includes a tag system, where voters can optionally explain why they rated a note a particular way. The system has a feature called *Tag Outlier Filtering*, which increases the intercept required for a note to be broadly shown to $0.5$ (instead of $0.4$) if a note receives a high volume of "Not Helpful" tags, such as "Sources not included or unreliable". Such a guardrail might lead us to overestimate MRS, as an adversary could get away with fewer votes if they optimize for tags.

## D. Attack Cost Model Calculations

This appendix provides justification for the cost estimates in Section 5.2.

**Model Training ($c_m \approx \$0$).** Embedding all notes in the dataset via Voyage AI (around 110M tokens) costs under $20 at $0.18/million tokens, but their free tier (200M free tokens) covers the entire dataset. Other CPU/GPU compute costs are negligible. Thus the total cost is effectively $0.

**Account Maintenance ($c_a \approx \$3$/month).** In addition to requiring accounts to be at least 6 months old before becoming eligible raters, Community Notes requires verified phone numbers from trusted carriers. Prepaid plans in the US (e.g., Ultra Mobile) cost $3/month per number.

**Vote Casting ($c_v \approx \$0.05$).** At 10 seconds per vote and $16.90/hour (CA minimum wage): $c_v = (10/3600) \times 16.90 \approx \$0.05$. Automation would reduce this to near-zero.

**Summary.**

| Term | Component | Estimate | Notes |
|------|-----------|----------|-------|
| $c_m$ | Model training | $\approx\$0$ | Due to free API tiers and minimal CPU/GPU cost |
| $c_a$ | Account maintenance | \$3/month | Phone carrier prices dominate cost |
| $c_v$ | Vote casting | \$0.05 | Negligible, based on CA minimum wage |
| $n_v$ | Votes required | 5-15 typical | Higher for higher visibility notes |
| $n_a$ | Accounts per vote | 1 | Anyone can vote on any note |

## E. Cost Model Justification

This appendix expands on the cost model in Section 5.2 by explicitly stating the assumptions on which it rests, and the sense in which it is sufficient for that purpose.

This cost model relies on three assumptions, each of which determines a specific aspect of the model. We state and justify each one below.

**(A1) Additive decomposition into independent phases.** The attack consists of operationally independent phases—model training, account acquisition, and vote casting—and the total cost is the sum of each phase's cost. This assumption determines both which terms appear and the model's functional form. These phases involve separate procurement channels (compute providers, phone carriers, manual labor) with no clearly shared infrastructure. This modular cost structure is well-established in the adversarial ML literature: for instance, Huang et al. (2025) decompose leaderboard manipulation cost into three similar categories (detector training, account creation, action execution). Further, Kireev et al. (2023) explicitly adopt modular costs as a named assumption, and Lowd & Meek (2005) define adversarial cost as a weighted sum of independent modifications.

This assumption might be violated if phase costs were coupled: for instance, investing more in model training ($c_m$) improves prediction of note parameters, enabling more precise account positioning and potentially reducing the number of votes required ($n_v$) in practice. In our setting, this interaction is negligible because $c_m$ is already near zero once prediction quality is sufficiently high, but it illustrates a dependence the additive form does not capture.

**(A2) Linearity.** Per-unit costs ($c_a, c_v$) are approximately constant over the modeled regime of 5–15 votes and dozens of accounts, leading to a linear scaling in cost. The justification is empirical: prepaid phone plans carry fixed monthly prices, and manual voting incurs approximately constant per-unit labor cost at this scale. Linear scaling of identities with resources is well established in the Sybil attack literature (Douceur, 2002; Wagman & Conitzer, 2014). This assumption might be violated at a larger scale, where bulk discounts on phone numbers could make $c_a$ sub-linear, while increased detection risk from coordinated activity could introduce super-linear costs to evade detection.

**(A3) Anonymity.** The attacker uses disposable accounts, so no reputational cost term is required. This is grounded in the current operations of these systems: for example, in X's Community Notes, any account with a verified phone number and six months of activity can become a contributor, and all contributions are displayed under auto-generated aliases rather than real usernames. This assumption would be violated if the system required verified real-world identity to participate, though this would conflict with the system's core commitment to anonymous contribution, intended to prevent retaliation against fact-checkers.

Under assumptions A1–A3, the model is sufficient for its stated purpose: it identifies account maintenance as the dominant cost component (orders of magnitude above model training and vote casting), and this determines which mitigations might be most effective. Importantly, the ranking is robust to moderate violations of A1–A2, as account costs remain the largest

term even under bulk discounts or mild interaction effects. Further, since total cost scales with $n_v$ (the MRS), the cost model connects the per-note vulnerability analysis in Section 5.1—which identifies precisely which notes are cheaply manipulable—to system-level mitigation design.

## F. Training Details

**Data.**  Our inputs are note text from the Community Notes dataset. Our target labels are the note factors and note intercepts, which are obtained by running the open source code for X's production Community Notes system on the complete dataset. It is worth noting that in practice, we observe that the inferred parameters are fairly stable across runs. For some theoretical intuition, after fixing the bias terms in the objective (Eq. 2), the only non-identifiability of the factors is a global scaling, $\langle f_u, f_n \rangle = \langle c \cdot f_u, \frac{1}{c} \cdot f_n \rangle$; however, with the regularization terms $\lambda_f (\|f_u\|^2 + \|f_n\|^2)$ in the objective, the scaling ambiguity is essentially removed and only a sign ambiguity persists ($\langle f_u, f_n \rangle = \langle -f_u, -f_n \rangle$), under which the predicted rating is invariant. We filter to notes with at least 26 ratings to ensure that the computed factors and intercepts are sufficiently reliable; this amounts to about $50\%$ of all notes on the platform. These notes span the entire time range (Jan 2021 to Jan 2025). We do not filter by any particular note status; thus, even notes that are not displayed publicly on the platform, but entered the rating phase (and achieved a "Not Helpful" or "Needs More Ratings" status) are included. We then embed note text with the `voyage-3-large` model ($d = 1024$) before feeding it into the MLP that we train.

**Splits and preprocessing.**  To account for temporal leakage, we use data from Jan 2021 till Sep 2024 as the training set ($n_{train} = 426535$) and data from Nov 2024 to Jan 2025 as the test set ($n_{test} = 114976$). This amounts to a rough 80/20 split. We hold out 10% of the training set as a validation set for our hyperparameter sweeps.

**Hyperparameters.**  We sweep learning rate $\{0.005, 0.001, 0.0005, 0.0001\}$, weight decay $\{0.01, 0.001, 0.0001, 0.00005\}$, hidden dimensions $\{[1024, 512, 128, 32], [512, 256, 128], [512, 128, 32], [256, 128], [128, 64], [64, 32]\}$, dropout $\{0.2, 0.3, 0.4\}$, and batch size $\{32, 128\}$. Each configuration is trained for up to 100 epochs with early stopping.

