# OpenReview forum: "Gaming Consensus: Coordinated Manipulation in Crowdsourced Fact-Checking"
_ICML.cc/2026/Conference — ICML 2026 regular_

### Official Review · Reviewer_aH5v · 2026-03-12

**Soundness:** 3
**Presentation:** 2
**Significance:** 3
**Originality:** 2
**Overall Recommendation:** 4
**Confidence:** 4

**Summary:**

The paper discusses an attack on the community notes feature, a crowd-sourced means of fact-checking in abundant use on social media platforms today. These systems use a scoring algorithm that promotes consensus among a diverse user base who do not agree with one another consistently on many issues. For this purpose, a recommender-system-esque algorithm is used for user feedback aggregation, which forms the main attack surface used by the authors. The attack involves the creation of multiple user accounts with optimized historical voting patterns to ensure they end up in support (e.g., by voting against one another on other notes), and then voting unanimously on the target note. This, of course, requires solving the recommender system matrix factorization problem efficiently. This is a key challenge given that it is dependent on the note parameters that need to be predicted. Here, an embedding model is used in combination with an MLP to regress the scoring function parameters and ensure the intercept (i.e., baseline support) for the note is above a certain threshold.  Authors also discuss mitigation strategies for the attack: in particular, by proposing a cost model for the full attack and running a comparative estimate for the cost of manipulating votes and accounts.

**Compliance With Llm Reviewing Policy:**

Affirmed.

**Final Justification:**

I think the follow up discussion does answer a key subset of my concerns—at least the ones that are possibly coverable in a rebuttal.

This is an attack paper, and arguably a rather simple attack. In the security community we do not begrudge an attack for its simplicity when it works. Therefore, despite the fact that I think the paper could have done much more with its theoretical formulation and connection to recommender systems; I think the treatment is sufficient for a first work. I vote for acceptance.

**Key Questions For Authors:**

- Q1. Have you considered how your approach connects to shilling (sybil) attacks against recommender systems?
- Q2. In Figure 4, why is there a jump at $f_u = 0.1$? The distribution otherwise looks uniform, which was your goal.
- Q3. Can you clarify what you mean in Line 224 by "gold truth values for fn and in obtained from running the open source codebase for the production Community Notes system."?

**Limitations:**

Yes. Limitations are discussed rather well.

**Strengths And Weaknesses:**

Overall, I think this is a good empirical paper. The problem is well-argued for and timely, given the prevalence of misinformation on social media platforms. Having said that, I think the writing is rather uneven in terms of technical detail. Also, there is an unfortunate effort to hide any theory in the appendix. This is the determinant of the paper. Also, in general, I think much more could have been done with the theoretical connection to recommender systems.

- **S1. The topic is timely.** Community notes are one of the only remaining (quasi-)fact-checking tools that remain in use. Their crowd-sourced model is clearly vulnerable to coordinated action.

- **S2. The paper features a thoughtful limitations section.** I think this is a big plus. Any modeling work is wrong at some level, but that makes it more important to specify where and when the model fails.

- **W1. Too many of the mathematical derivations are relegated to the appendix.** For example, I found the derivation of the Optimal Single Rating Injection pretty hard to read. What is the regression problem? What is the Gram matrix that you suddenly reference?  Similarly, in Line 280, the reader is asked to visualize a (arguably simple) mathematical result. Why not have the visualization in the paper? The reader is then promptly asked to follow the derivation in Appendix A. I understand that the formalism is introduced in Appendix A, but this is not good academic writing.

- **W2. Uneven level of detail.** The level of detail is uneven throughout the paper. Until Page 4, there can be too much redundancy. For example, the paper spends too much time on separating Phase 1 and Phase 2, while Phase 2 in Section 3 follows rather trivially once the answer to Phase 1 is discovered. I also disliked how the paper constantly teased certain results without actually presenting their justification. For example, the expression "rating a note as 'Not Helpful' can sometimes increase its intercept." is repeated at least 3-4 times before the paper gets to explain why this is the case. Yet, simple but informative derivations are dropped from 5.1, which hurts readability. (see previous point)

- **W3. The cost model in Section 5.2 is ad hoc.** The paper suggests that Eq. 6 is a "Comprehensive" cost model for the manipulation difficulty. I think this is an overstatement. First, the cost does not need to be linear. Second, different users might have different risk-aversion levels. Third, reputational damage and long-term dynamics of the manipulation of votes are not considered. I don't think the paper offers a compelling reason why considering these factors is **sufficient**.

    \
    To be clear, this is not a knock on simplified models. But such a model needs a purpose to exist. The paper currently enumerates cost components and their mitigation, only to finish by providing a comparative cost analysis for a single-vote, with the conclusion being that the dominant cost is account maintenance. I think this is a lost opportunity to incorporate cost (utility) analysis in the models developed in Section 5, to assess a proper utility function for the attacker. **One which allowed us to reason about what notes, under what conditions, are specially vulnerable to manipulation rather cheaply.**
- **W4. The recommender system connection is underexplored.**   I think the connection to recommender systems is heavily underutilized here. There is so much literature on Byzantine failures of these systems, like shilling (sybil) attacks, that would have applied here. I would have been interested to know if deferences against those would help here. More importantly, it would have provided reasonable baseline results to compare the method to.

- Minor

	- Most plot fonts are too small which degrade readability.

---

> ### Author Rebuttal · Authors · 2026-03-31
>
> Thank you for such a thoughtful review! We are glad you found our work to be a good empirical paper and found the problem well-argued for and timely. We address your questions and comments below:
>
> [W1, W2] Thanks for the constructive writing feedback! In the revised manuscript, we will incorporate your feedback through various changes, such as including a visualization of the result in Line 280 and moving more details about the derivations in Section 5.1 from the Appendix into the main paper to improve the flow and evenness of detail.
>
> [W3] We agree that describing our cost model as “comprehensive” could be read as an overstatement. We will revise the wording to avoid this implication.
>
> Our goal with the cost model is to provide a simple, interpretable abstraction of the key levers that influence manipulation difficulty, rather than a fully general model. In that spirit, we adopt linearity as a simplifying assumption. We also expect manipulation to be dominated by disposable accounts (which are anonymous) in our setting, which may limit reputational concerns. We agree that richer formulations (non-linear models, user models with risk tolerance etc.) are possible and certainly worth exploring in future work.
>
> We will revise the text to better position this model as a simplified characterization with the goal of identifying key mitigation surfaces and explicitly acknowledge these limitations.
>
> [W4, Q1] We thank the reviewer for the helpful suggestion, and we will incorporate a fuller discussion of this connection in the revised related works section.
>
> As briefly hinted in Section 2, Community Notes and matrix factorization-based recommender systems are similar primarily at the modeling level: both learn latent representations of users and items based on a sparse user-item matrix. The subsequent goals of the system are largely different. In the recommender setting, matrix factorization is predominantly used for personalized prediction or ranking based on relative positions in latent space; however, Community Notes uses the note intercept as a single global score of note quality, departing from the traditional use of MF in recommender systems.
>
> Consequently, the attack objectives differ as well. Classic recommender-system shilling attacks aim to push a target item, nuke a target item, or, more generally, compromise recommendation integrity by injecting fake user profiles, including through well-known heuristic attack families such as random, average, bandwagon, and segment attacks (Lam and Riedl, 2004; Mobasher et al., 2005; Mobasher et al., 2007). In contrast, the natural adversarial objective in Community Notes is less so to “promote one item” than cultivate accounts that can fabricate diverse agreement and eventually potentially manipulate many arbitrary notes.
>
> Even at the mechanism level, Community Notes presents an online problem where unlike classical recommender systems, users cannot inject arbitrary rows into the matrix (e.g. rate all available items on a platform), but can only add ratings to “new items” to manipulate the system successfully. Moreover, attacks in modern recommender settings, such as PoisonRec (Song et al., 2020), look quite different because they operate in an implicit-feedback setting rather than one with binary votes. Finally, it is worth noting that Community Notes provides an unusually white-box setting for the attacker, since both the scoring code and complete data are published daily, whereas many popular large-scale recommender systems (e.g., Netflix, Amazon) are comparatively more black-box.
>
> [Q2] This is merely an artifact of noise in our prediction model and discretization from bucketing in our histogram in the finite-sample regime rather than a meaningful deviation from uniformity.
>
> [Q3] We are referring to “true” values of $f_n$ and $i_n$ as inferred by the actual production Community Notes M system. Note that these values can be obtained for past notes by the attacker (and us) by running the open source codebase on the full data, which is publicly released. We use these values as labels when training our prediction model. We realize that the term “gold truth” is non-standard, and we will replace it with “ground truth” for clarity.
>
> We hope the reviewer finds our answers helpful. Please do not hesitate to let us know if any questions or concerns remain. Thank you again for your time and consideration.
>
> References:
> - Lam, S. K., and Riedl, J. (2004). Shilling Recommender Systems for Fun and Profit.
> - Mobasher, B., Burke, R., Bhaumik, R., and Williams, C. (2005). Effective Attack Models for Shilling Item-Based Collaborative Filtering Systems.
> - Mobasher, B., Burke, R., Bhaumik, R., and Williams, C. (2007). Toward Trustworthy Recommender Systems: An Analysis of Attack Models and Algorithm Robustness.
> - Song, J., Li, Z., Hu, Z., Wu, Y., Li, Z., Li, J., and Gao, J. (2020). PoisonRec: An Adaptive Data Poisoning Framework for Attacking Black-Box Recommender Systems.

---

> > ### Author Rebuttal · Reviewer_aH5v · 2026-04-01
> >
> > Thank you for the rebuttal. I especially liked the discussion in "[W4, Q1]." As a matter of establishing positionality, I am a security researcher. To me, this discussion of the threat model should have been featured prominently in the paper.
> >
> > I still do not agree with the cost modeling decisions. I do not refute your arguments that "manipulation to be dominated by disposable accounts (which are anonymous) in our setting, which may limit reputational concerns." However, this is a modeling assumption rather than a result—at least this is how it is presented in the paper.
> >
> > I think a good model should at least attempt to justify itself axiomatically. It should state the purpose of the model and argue for the sufficiency of the axioms for that purpose, and discuss under what conditions these assumptions may fail.
> >
> > Thank you for answering my questions. "gold-tasks" are known in the crowd-sourcing context, and it is a better description that ground truth. I would keep the phrasing and instead explain it. (you do not, in fact, observe ground truth in such contexts)
> >
> > All and all, I think I am inclined to keep my score. If there's a discussion I'd still argue for acceptance.
> >
> > I wish you all the best

---

> > > ### Author Response · Authors · 2026-04-07
> > >
> > > Thank you for your thoughtful response! We are glad you found our extended discussion on the connection to recommender systems helpful; we will feature this more prominently in the updated manuscript.
> > >
> > > We also appreciate the constructive feedback regarding the cost model section of the paper, and we agree that the model’s purpose and assumptions can be more explicitly stated and justified. To further improve the paper, we will revise this section with the discussion below.
> > >
> > > **Purpose:** As we clarified in our earlier response, the cost model is intended as a simple, interpretable abstraction of the key levers that influence manipulation difficulty, rather than a fully general attacker utility model. We will revise Section 5.2 to explicitly scope this purpose: to identify which cost components dominate and to enable comparison of mitigations.
> > >
> > > **Assumptions:** The cost model relies on three assumptions, each of which determines a specific aspect of the model. We state and justify each one below.
> > >
> > > **(A1) Additive decomposition into independent phases:** The attack consists of operationally independent phases — model training, account acquisition, and vote casting — and the total cost is the sum of each phase's cost. This assumption determines both which terms appear and the model's functional form. These phases involve separate procurement channels (compute providers, phone carriers, manual labor) with no clearly shared infrastructure. This modular cost structure is well-established in the adversarial ML literature: for instance, Huang et al. [1] decompose leaderboard manipulation cost into three similar categories (detector training, account creation, action execution). Further, Kireev et al. [2] explicitly adopt modular costs as a named assumption, and Lowd & Meek [3] define adversarial cost as a weighted sum of independent modifications.
> > >
> > > This assumption might be violated if phase costs were coupled: for instance, investing more in model training ($c_m$) improves prediction of note parameters, enabling more precise account positioning and potentially reducing the number of votes required ($n_v$) in practice. In our setting, this interaction is negligible because $c_m$ is already near zero once prediction quality is sufficiently high, but it illustrates a dependence the additive form does not capture.
> > >
> > > **(A2) Linearity:** Per-unit costs ($c_a, c_v$) are approximately constant over the modeled regime of 5–15 votes and dozens of accounts, leading to a linear scaling in cost. The justification is empirical: prepaid phone plans carry fixed monthly prices, and manual voting incurs approximately constant per-unit labor cost at this scale. Linear scaling of identities with resources is well established in the Sybil attack literature (Douceur [4],  Wagman & Conitzer [5]). This assumption might be violated at a larger scale, where bulk discounts on phone numbers could make $c_a$ sub-linear, while increased detection risk from coordinated activity could introduce super-linear costs to evade detection.
> > >
> > > **(A3) Anonymity:** The attacker uses disposable accounts, so no reputational cost term is required. This is grounded in how Community Notes operates today: any account with a verified phone number and six months of activity can become a contributor, and all contributions are displayed under auto-generated aliases rather than real usernames. This assumption would be violated if the system required verified real-world identity to participate, though this would conflict with the system's core commitment to anonymous contribution, intended to prevent retaliation against fact-checkers.
> > >
> > > **Sufficiency:** Under assumptions A1–A3, the model is sufficient for its stated purpose: it identifies account maintenance as the dominant cost component (orders of magnitude above model training and vote casting), and this determines which mitigations might be most effective. Importantly, the ranking is robust to moderate violations of A1–A2, as account costs remain the largest term even under bulk discounts or mild interaction effects. Further, since total cost scales with $n_v$ (the MRS), the cost model connects the per-note vulnerability analysis in Section 5.1 — which identifies *precisely which notes are cheaply manipulable* — to system-level mitigation design.
> > >
> > > We hope this discussion helps strengthen the presentation of the cost model in our paper.
> > >
> > > Thank you again for your engagement and your overall positive assessment of our paper, and please let us know if you have any other suggestions to improve the final version of the paper!
> > >
> > > --
> > >
> > >
> > > [1] Huang et al., Exploring and Mitigating Adversarial Manipulation of Voting-Based Leaderboards, ICML 2025
> > >
> > > [2] Kireev et al., Adversarial Robustness for Tabular Data through Cost and Utility Awareness, NDSS 2023
> > >
> > > [3] Lowd & Meek, Adversarial Learning, KDD 2005
> > >
> > > [4] Douceur, The Sybil Attack, IPTPS 2002
> > >
> > > [5] Wagman & Conitzer, False-Name-Proof Voting with Costs over Two Alternatives, IJGT 2014

---

### Official Review · Reviewer_8UxN · 2026-03-13

**Soundness:** 3
**Presentation:** 3
**Significance:** 3
**Originality:** 2
**Overall Recommendation:** 4
**Confidence:** 3

**Summary:**

This paper consider the robustness of crowdsourced fact-checking systems,  focusing on algorithms used in Community Notes style mechanisms deployed by major platforms such as X, Meta, TikTok, and Google.  These systems use a matrix-factorization based scoring algorithm.  The paper show that this mechanism can be strategically manipulated by coordinated agents. By carefully choosing how they vote, attackers can artificially generate diverse agreement in the latent factor space. The paper proposes an attack strategy where a small coordinated group manipulates the consensus process.  Experiments  indicate that, in some cases, fewer than ten coordinated users may be sufficient to influence which notes are surfaced. The authors also introduce a cost model for such attacks and discuss possible mitigations.

**Compliance With Llm Reviewing Policy:**

Affirmed.

**Final Justification:**

After careful consideration, and taking into account the other reviews, I believe the problem addressed in the paper is novel. I therefore support acceptance and have revised my score accordingly.

**Key Questions For Authors:**

Please see the weaknesses noted above.

**Limitations:**

Yes

**Strengths And Weaknesses:**

Strengths:

1. The paper is well written and studied problem is interesting.

2. Provides a scoring and attack model.

3. Experiments on real data and discuss possible mitigation strategy.

Weaknesses:
1. The attack assumes that adversaries can coordinate voting patterns strategically but it is unclear:
a) the difficulty of discovering the latent structure and b)how attackers would learn or estimate the matrix factorization model.

2. How robust is the attack if the platform periodically re-trains  the matrix factorization model?

3.  How sensitive is the Phase 1 strategy to the choice of embedding model?
 Could the platform detect Phase 1 behavior by looking for users whose rating patterns appear
 too optimized for achieving specific latent factor positions?

---

> ### Author Rebuttal · Authors · 2026-03-31
>
> Thank you for your review! We address your questions below:
>
> 1. For historical notes, the attacker can reconstruct the relevant latent parameters by running the public, open-source scoring code on the public ratings data. It is worth noting that such commitment to transparency and open data is vital for a crowd-sourced fact-checking platform to succeed in practice (“The program is built on transparency: all contributions are published daily, and our ranking algorithm can be inspected by anyone” - https://communitynotes.x.com/guide/en/about/introduction). The prediction model is introduced because the Community Notes data is published with a 48-hour lag (see footnote 7), so an attacker needs to estimate note parameters immediately to enable Phase 1 strategic voting without delay.
>
> 2. In production systems like X’s Community Notes, the matrix factorization model is periodically re-trained, but in practice, we observe that the inferred parameters are fairly stable across runs. Moreover, X’s implementation contains logic to help with stability (e.g. re-training if parameters drifted too much from previous runs, perhaps due to unlucky initialization). For some theoretical intuition, after fixing the bias terms in Eq.2, the only non-identifiability is $<f_u, f_n>=<c \cdot f_u, \frac{1}{c} \cdot f_n>$. However, with the regularization $ \lambda_f \cdot (|f_u|^2 + |f_n|^2),$ the scaling ambiguity is essentially removed and only the sign ambiguity persists ($<f_u, f_n>=<- f_u, - f_n>)$, and the predicted rating (Eq. 1) is invariant under this sign change. We will make sure to clarify this in the updated manuscript.
>
> 3. This is a great question! We found that we did need a strong embedding model (voyage-large-3 in our case) to achieve our results; for instance, merely using an arbitrary embedding model from HF did not suffice. Importantly, however, the prediction models we use are not perfect  (see Figure 3), but despite this noise, the resulting accounts still span the target factor range in Figure 4, as required  to successfully execute Phase 1.  Therefore, detecting users whose rating histories appear unusually optimized is a plausible mitigation, but the detector would need to distinguish adversarial coordination from ordinary noise. Further, the adversary might also be able to evade detection by adding noise to otherwise “perfect predictions”.
>
>
> We hope this clarifies the reviewer’s concerns. Please let us know if we can help resolve any other questions you might have.

---

> > ### Author Rebuttal · Reviewer_8UxN · 2026-04-02
> >
> > Thank you for the response. My questions have been addressed by the authors. Overall, I find the paper is good  and would like to maintain my original score.

---

> > > ### Author Response · Authors · 2026-04-07
> > >
> > > Thank you for the follow-up and the positive assessment of our paper! We're glad to have addressed your questions. Since the acknowledgment mentions remaining follow-up questions, we'd be very happy to address any lingering concerns. In particular, if there is a remaining issue affecting your final recommendation, it would be helpful for us to know so that we can address it as directly as possible in the revision. If not, we would be very grateful if you could consider revising your score to reflect your positive assessment of our paper.

---

### Official Review · Reviewer_1eUq · 2026-03-19

**Soundness:** 3
**Presentation:** 3
**Significance:** 3
**Originality:** 3
**Overall Recommendation:** 4
**Confidence:** 3

**Summary:**

This paper evaluate the security of crowdsourced factchecking systems used by big platforms like X and Meta. These systems use a "bridging" algorithm based on matrix factorization to filter out misinformation by requiring support from diverse groups instead of just a majority vote. The core model is defined as:$$\hat{r}_{un} = \mu + i_u + i_n + f_u \cdot f_n$$Even though this is supposed to be robust against "brigading," the authors show a coordinated attack where users can fake diversity in the latent space to trick the scoring. They first "train" their accounts to look like they have different viewpoints by voting on old notes, then they all support a target note to push it's intercept $i_n$ above the threshold.The main findings of this work are quite surprising:Small scale attack: For many notes, it only takes a small group (less than 10 accounts) to manipulate the status and surface arbitrary content.Algorithmic flaw: The paper proves a weird property where rating something as "Not Helpful" can actually increase the helpfulness score of a note under certain conditions.Real world impact: They built a cost model showing the attack is cheap (around $30 per note) and worked with X's team to deploy mitigations before publishing.

**Compliance With Llm Reviewing Policy:**

Affirmed.

**Key Questions For Authors:**

Here are 4 questions for the authors:
1. Robustness to Noise: If the MLP's predictions for note parameters ($f_n$ and $i_n$) are slightly inaccurate, does the attack still succeed with <10 votes? This would clarify how much precision an attacker really needs.
2. Higher Dimensions: If the system increases the factor space from $d=1$ to $d=5$, does the number of accounts needed to "position" themselves increase significantly? This helps evaluate if increasing dimensionality is a strong enough defense.
3. Spam Protection: For a note that is obviously low-quality or spam, is the 5-15 vote threshold still enough to make it "Helpful"? I want to know if the system has any basic content-based resistance.
4. Live Correction: Since this is a static data study, how long would a manipulated note stay visible before real users likely downvote it and correct the score? This would help me understand the actual "window of harm" for this attack.

**Limitations:**

Yes, the authors have included a thorough discussion of limitations in Section 6.1 and a dedicated impact statement. They are transparent about using a static data snapshot rather than a live system and the use of a greedy algorithm for their calculations.

**Strengths And Weaknesses:**

StrengthsSoundness: The paper is technically very solid, especially with the mathematical derivation in the Appendix where they use the Sherman-Morrison formula for rank-one updates to the Gram matrix. They don't just speculate; they provide a closed-form solution for the optimal single rating injection, which makes the attack very convincing. Also, the use of real production data from X (Jan 2021 - Jan 2025) gives the empirical results a lot of weight.Significance: This work has huge practical value. Since platforms like TikTok and Google are moving toward this "bridging" model for fact-checking, knowing these systems can be gamed with just ~10 accounts is a big deal. The fact that X's team already implemented mitigations based on this research proves its real-world impact.Originality: Most previous work on manipulating voting systems looked at simple majority rules, but this is the first rigorous look at the "bridging" matrix factorization logic. The insight that "user type" is determined solely by voting history—and thus can be "faked"—is a very clever perspective.Presentation: The overall narrative is easy to follow. Even though the math gets heavy in the Appendix, the main text does a good job of explaining the two-phase attack (Phase 1 for positioning, Phase 2 for boosting) without getting lost in the weeds.WeaknessesSoundness (Static vs. Live): One limitation is that the experiments were done on a static snapshot of data. We don't really know how the "live" system would react if real users saw these manipulated notes and started downvoting them immediately.Soundness (Greedy Approximation): To calculate the Manipulation Resistance Score (MRS), the authors used a greedy algorithm. As they admit themselves, this might actually understate how easy the attack is, because a global optimization might find an even smaller number $k$ of accounts needed.Originality (Simple MLP): The model used to predict note parameters from text is a fairly basic MLP and ignores URLs or the actual post being annotated. Including that context would probably make the attack much more accurate, but it's not a dealbreaker for the paper's main point.Significance (Dimension d=1): The analysis mostly focuses on the $d=1$ case used in production. While they discuss higher dimensions in the mitigations, it's not totally clear if the attack stays as cheap or efficient if the system moves to much higher dimensional latent spaces.

---

> ### Author Rebuttal · Authors · 2026-03-31
>
> Thank you for your encouraging review! We are glad you found our work technically very solid and believe it has huge practical value. We address your key questions below:
>
> 1. Yes, our Phase 1 experiment already operates in the noisy setting where the adversarial accounts choose ratings using predicted note parameters from Section 4.2 rather than oracle values. In fact, as seen in Figure 3, the resulting accounts still span the target factor range in Figure 4. Thus, the predictions have sufficiently low noise / high signal to make the attack successful.
>
> 2. Yes, as discussed in Section 5.2 (line 374), we would expect that increasing the dimension would significantly increase the number of adversarial votes (and hence accounts) required to execute the attack. However, as also noted in Section 5.2, higher-dimensional factors are not a feasible mitigation to this problem and would be undesirable in practice, as “the algorithm may learn to bridge across dimensions that do not correspond to meaningful ideological divisions, such as writing style, tone, or use of humor. In practice, $d=1$ has proven effective because it captures the dominant axis of disagreement, typically corresponding to political orientation.”
>
> 3. No, for a note that is obviously low-quality or spam, the 5-15 vote threshold is not typically enough to make it "Helpful". Such notes typically have a very low note intercept to begin with (corresponding to the “points on the left of the plots” in Figure 5), and would generally require a much higher number of votes to be manipulated. It is worth noting, however, that we do not view notes that are obviously spam as our primary threat vector; see the discussion in Section 6 for more details.
>
> 4. This is a great question for future work! It is worth noting that production implementations such as X’s Community Notes have features such as ‘CRH inertia’, which add resistance for a note that is marked ‘Helpful’ to lose its helpfulness status; however, we do not wish to speculate on the feedback dynamics as it is beyond the scope of this work. See Section 6.1 for additional discussion on the same.
>
> We hope the reviewer finds our answers helpful. Please do not hesitate to let us know if any questions or concerns remain. Thank you again for your time and consideration.

---

### Official Review · Reviewer_QmtJ · 2026-03-19

**Soundness:** 2
**Presentation:** 3
**Significance:** 2
**Originality:** 2
**Overall Recommendation:** 3
**Confidence:** 4

**Summary:**

This paper proves that a non-peer reviewed approach to content moderation a la crowdfunding, with a single dimension d=1, is easily vulnerable to targeted placement. The result are at the same time useful (given the poor state of social media content moderation) and somewhat very expected.

**Compliance With Llm Reviewing Policy:**

Affirmed.

**Final Justification:**

I have addressed additional information from the rebuttal.

I maintain that, technically speaking, this paper's conclusion is too expected to grant publication (the attack is so simply and a system as badly designed as community notes, and not peer reviewed, even if it's deployed, is simply not credible enough), but I think my personal view should not be blocking the paper to be published if other reviewers, after considering my review, think the contribution of interest to the community.

**Key Questions For Authors:**

Should article pointing the flaw of "bad research" be a subject for more research articles? I would say, conditionally, yes. There has been a lot of papers pointing to limitations of previous works, they tend to fit in a few categories:
1) overlooked aspects: A paper shows that a technique proposed by paper a has side effects that were difficult to spot or became apparent when introducing new techniques.
2) as stepping stone: A paper shows that a technique proposed by a paper is limited and proposes an advance to remedy that.
3) as a cautionary tale: A paper shows that a technique is flawed and that previous report of its results are misleading.

This article appears to be in the 3rd category (with a few short forrays in 2) : It does not introduce a new dimension about birdwatch (the original report indicates that it introduces its specific emphasizes on intercept to take some manipulation into account). Basically we have not learned any new dimensions of metrics or efficiency. After showing that it's pretty trivial to optimize for an attack when d=1 with access to coefficient, it proves that it can be done. It introduces some advice about mitigation strategy but those tend not to be real "research contribution" and more discussion item as if the paper was a sort of position paper. In particular none of those are demonstrated to offer a robust and solid mix.

Because the problem is posed with d=1, the technical problem to solve is very contrived. Putting it differently, all the difficulty in social network manipulation is that there is a large dimensional space and that your accounts need to be at the same time relevant to a target (for instance, to create from scratch or amplify a grass root movement) but without becoming so narrow that it is easy to detect as coordinated. Those are properties typically obtained for large d. Appearing "diverse" in d=1 when you know the coefficient is quite simply computing a multiplication.

Local minor comments:

p.1 "sometimes at the expense of getting rid" I guess you mean either "sometimes to get rid" or "at the expense of hiring"

p.2 "carefully designed voting strategy" I am sorry but I feel this is not accurate: picking a random subset and reading the coefficient to pretend to be a point in a unidimensional space is not "careful design". But in a way it makes the result stronger. You could say that an elementary attack succeed. I feel that makes the story clearer and the point stronger that Community Notes is vastly overselling its strength.

Perhaps the paper could explore if community notes was somewhat less trivial (d>1) and whether some of those attacks could be extended, making the result more significant.

(NB this point was entirely address in rebuttal, only leaving for completeness
p.3 "publishes code" that's ok but it does not publish lambda i and lambda f .)

p.4 predicting note parameters from text. I was confused by that since it seems to go against the grain: first, it would not necessarily be public how text would contribute (and makes the crowdsourcing actually more robust), and it was clear before that f_i and f_n are computed from optimization of votes, not from text. Remember that d=1 so what are we even talking about here?

p.6 "d=1" I have to be honest that when reading d=1 after reading five pages!! I sort of fell from my chair. I was in my head trying to figure the possible challenge? and also wondering how come fig. 4 and 5 present the factor f as a scalar. And then the big reveal more than half way through the paper that this original design is incredibly limited. It's such a strong assumption that, either you put it in the front as an incredible poor choice (and show that nevertheless your attack would apply for larger d). Or you at least warn from the beginning that "latent space" everywhere here means [-0.5;0.5]. That would require some explanation why we should even consider such an extreme choice to be of scientific interest for a crowdsourcing method?!
(this point was partially addressed in rebuttal and I will have a look at the final version to check)

**Limitations:**

I think the authors are probably motivated by the fact that communitynote is, in the grand scheme of things, a less than ideal solution. It belongs to the bag of "replacing teacher with AI" and "let a robot be a companion to solve isolation" kind of thing. It poorly disguised the intention to basically cut down cost and worsen service. In that sense, I feel their paper is useful as it at least state very clearly why this is so flawed, but let's not forget that noone in our research community ever thought it was good (the solution is not published or peer reviewed or considered scientifically sound).

One minor concerns is that section 5 with "mitigation" paragraphs read like "more ways to continue to justify a bad idea". I don't think it's unethical (it's reasonable to point out to what's marginally better) but perhaps the authors may want to consider whether that may simply give more arguments to continue pretending that crowd sourced moderation is sustainable.

**Strengths And Weaknesses:**

Strengths are:
- To the extent that a crowdfunding solution that is doing the absolute bare minimum to avoid collusion is being implemented and considered a contender (by whom?), that paper confirms that it is easily susceptible to manipulation.

Weaknesses are:
- It's hard to see a technical contribution here. Crowd sourcing being vulnerable to adversary attacks is very well known, the method are straightforward (and they should be, since the original design was so obviously flawed, I'm not criticizing it).
- One could conceive potentially some interesting challenge for large d (since it might be difficult for an adversary to find documents to create relevant diversity about). But the d=1 assumption makes everything so narrow (you can even plot the "vector of factor"?!).
- Birdwatch has not been peer reviewed, it has been published entirely by a team of Twitter, and as pointed in the paper is basically a fig leaf to cost cutting by removing more expensive (and robust) content moderation.
NB: In rebuttal (and other reviewers) it remains true that birdwatch has been "cited" and compared, which makes it hard to entirely ignore today, giving more ground for this paper to get merit to point to its flaw.

---

> ### Author Rebuttal · Authors · 2026-03-31
>
> We thank the reviewer for their review. Before addressing specific comments and questions, we would like to clarify several points that underlie many of the concerns. Several of the concerns raised center on whether Community Notes is itself a worthwhile system to study, rather than on the correctness or significance of our analysis. These critiques seem directed at the design of the Community Notes system itself rather than the contributions of our paper. Our goal is not to advocate for Community Notes or endorse the original Birdwatch paper. Rather, we analyze a crowdsourced fact-checking system that is already deployed to millions of users and rapidly gaining widespread adoption (it is now deployed on X, Meta, TikTok, and YouTube), with the goal of identifying vulnerabilities in the system. As is good practice in security research, we also discuss potential mitigations to make such systems more robust.
>
> We now address specific points.
>
> - Relevance of Community Notes:
>     - The reviewer states that "noone in our research community ever thought it was good." We respectfully note that Community Notes has been studied in peer-reviewed work at PNAS (Slaughter et al., 2025), PNAS Nexus (Drolsbach et al., 2024), ACM Web Conference (De et al., 2025), ACL (Borenstein et al., 2025), and CHI (Lloyd et al., 2026), among others. A substantial and growing research community engages with this system.
> - Dimension $d=1$ of latent space:
>     - This is the production setting of Community Notes, not a simplifying assumption. Since the system is currently deployed in practice, affecting millions of users, showing a concrete attack in this setting is itself significant. We agree, however, that this point should be stated more prominently earlier in the paper—currently, it is stated on page 2 (footnote 5), not introduced on page 6 as suggested—but we will move it into the main text. We discuss increasing $d$ as a mitigation in Section 5.2 and explain why it is not straightforward: higher dimensions lead to bridging across non-ideological axes (writing style, tone), undermining the system's core purpose.
> - The necessity of predicting note parameters (factor and intercept) from text:
>     - The text-based prediction model is not part of the Community Notes scoring system; it is a component of our attack. Community Notes data is published with a 48-hour lag (see footnote 7), so an attacker needs to estimate note parameters immediately to enable Phase 1 strategic voting without delay.
> - “'publishes code' that's ok but it does not publish lambda i and lambda f .”
>     - The source code for Community Notes is fully public, and thus the regularization parameters $\lambda_i$ and $\lambda_f$ are specified in the code and fully known to the attacker (see Section 3).
>
> We hope the reviewer finds our answers helpful. Please do not hesitate to let us know if any questions or concerns remain. Thank you again for your time and consideration.

---

> > ### Author Rebuttal · Reviewer_QmtJ · 2026-04-03
> >
> > Thank you for taking the time to clarify multiple points of my review, including that I was wrongly assuming that two parameters were not public, and provide a set of papers confirming that community notes is (sadly) still not completely discredited in the eyes of current research. I do however feel that a subset of the paper included in your rebuttal mentions community nodes more as a design space (for instance, the idea of adding text next to the comment) rather than building upon it as if it was a satisfactory solution (as would, for instance, a paper totally endorsing its design and showing robustness).
> >
> > I have also noted that other reviewers do consider the application of community notes somehow established. It seems therefore useful to make it clear how flawed it is, which is what your paper is presenting.
> >
> > I think that paper that shows interests in the community should not be prevented to be published by the fact that some scores differ, so I'm happy to increase my score to a weak reject and allow other reviewers to promote/champion it for publication. I remain a bit baffled that fooling a unidimensional algorithm is considering a technical contribution, which is the main reservation and why I keep my score on the "negative" side, but I will not stand against the paper should the consensus be to accept it to be published.

---

> > > ### Author Response · Authors · 2026-04-07
> > >
> > > Thank you for your thoughtful follow-up! We are glad to have resolved several misunderstandings and to have addressed your key concerns about the significance and impact of Community Notes. While we understand that we differ in our assessment of the technical contribution, we’re encouraged by your recognition of the work’s relevance and appreciate your updated evaluation.
> > >
> > > Thank you again for your time and consideration.

---

### Official Review · Reviewer_U3f1 · 2026-03-24

**Soundness:** 3
**Presentation:** 3
**Significance:** 3
**Originality:** 3
**Overall Recommendation:** 4
**Confidence:** 3

**Summary:**

The authors present vulnerabilities in the crowdsourced fact-checking systems, Community Notes, which is designed to achieve diverse agreement for broader consensus regarding the quality of the note. The authors develop a 2-phase attack where an adversary can create numerous accounts to inject ratings on a given note to drive its rating to the desired value (specifically “Helpful”). This attack is empirically validated and studied on a publicly available real-world production-based dataset, accompanied by theoretical insights and cost of performing the attack. The results show that many notes require only 5-15 rating injections at a moderate cost.

**Compliance With Llm Reviewing Policy:**

Affirmed.

**Key Questions For Authors:**

1. Does the adversary know the value of $\tau$ for different categories of notes or do they set it to 0.4 for all notes as mentioned on Line 126, assuming that $\tau$ is a parameter disclosed in the open source code?
2. Would the same attack apply to rating a note as “Not Helpful”, perhaps with $i_n \lt \tau$?  This is important because misinformation is not only about making a note appear but also making the facts in notes disappear.
3. The experiments are done on X data based on its open-source code. Why were not these experiments done on other Community Notes dataset, if the claim is that these are open sourced and public datasets?
4. Regarding method: Why is $f_{target}$ uniformly spaced in range of [-0.5,0.5], why not [-1,1] as in Figure 2? If $f_a$ is to be learned next, how would choosing $f_{target}$ effect the learning of $f_a$? Why consider only 100 random notes? Would choosing i_target differently affect the attack? I understand that adversary chooses ratings on Line 264 using method in Line 275, please clarify, if yes why does method in Line 275 not consider 0.5 as rating (Line 278).

**Limitations:**

Yes

**Strengths And Weaknesses:**

Strengths:

1.	The authors address an important problem of detecting vulnerabilities in methods mitigating misinformation in social media. This drives the research towards building and strengthening such methods against manipulation to build trust.
2.	The authors provide adequate details, and the paper is well-motivated.
3.	The authors propose a simple yet effective attack to manipulate ratings that is theoretically justified via claims and proofs.
4.	The authors note that the paper presents complementary work relative to existing literature.
5.	The cost analysis of performing this attack is well discussed.

Weaknesses:

1. The proposed attack is shown only for changing the rating to “Helpfulness”
2. The proposed attack is heavily driven by equation 1, assuming that all methods, existing and new, will use the same scoring algorithm. This is under the assumption that all code and data will be made available in timely manner. Precisely, this is white-box attack with some known parameters such as $\mu$, $\tau$.
3. Learning different parameters in stages could be shown through a block diagram for clarity. For example, learning $f_a$ after positioning $f_{target}$ could be better described.
4. The effectiveness of the approach is based only one (but large real) dataset.
5. The performance of the attack on the diversity of notes and users (in terms of concepts) in the given timeline could be discussed.
6. Not a weakness, but authors can position their work in a much better way compared to existing works.

---

> ### Author Rebuttal · Authors · 2026-03-31
>
> Thank you for your constructive review! We are glad you find our paper well-motivated. We address your key questions below:
>
> [1] Yes, $\tau$ is typically set to 0.4, and is a parameter disclosed in the open source code and accompanying documentation. In rare cases, the threshold could be higher; see the discussion of “Tag Outlier Filtering” in our response to Question 2 below.
>
> [2] Yes, the same attack would apply to adversarially voting on a note that is currently rated “Helpful” to “take it down” and make it lose its helpful status. In terms of the optimal rating to inject, rather than maximizing $\Delta i_n$ (in Eq. 4), we would minimize it. It would still be a second-degree expression that can be easily optimized.
>
> In the camera-ready version, we will include the corresponding plot of the Manipulation Resistance Score (Figure 5), where we aim to lower the intercepts of notes with intercepts above 0.4. Once again, we observe that many notes can be brought down to below the 0.4 threshold with a handful of adversarial votes.
>
> It is worth emphasizing, however, that we need to be careful when interpreting these results, since we only study the attack on the core matrix factorization component. The production implementations of these matrix factorization systems often have some additional safeguards and features that make a direct extrapolation from our study to the exact number of votes required tricky. For instance, X’s implementation of Community Notes has a feature called ‘CRH inertia’, that requires that ‘Helpful’ notes drop below a note intercept threshold that is lower than the original threshold $\tau$ in order to lose their helpfulness status and stop being shown broadly. For instance, in some cases, a note that gained ‘Helpful’ status by reaching a note intercept of $\tau=0.40$ must drop below $\tau ’=0.39$ in order to lose its helpfulness status. Such a guardrail might lead us to underestimate MRS.
>
> On the other hand, X’s implementation of Community Notes also includes a tag system, where voters can optionally explain why they rated a note a particular way. The system has a feature called Tag Outlier Filtering, which increases the intercept required for a note to be broadly shown to $0.5$ (instead of $0.4$) if a note receives a high volume of “Not Helpful” tags, such as “Sources not included or unreliable”. Such a guardrail might lead us to overestimate MRS, as an adversary could get away with fewer votes if they optimize for tags.
>
> [3] At the time of writing, X’s Community Notes was the only publicly available dataset. Other companies, such as Meta and TikTok, are still in the initial phases of rolling out this feature (given they’ve only announced this shift within the last year), and are yet to publicly release datasets at the scale of X (which has supported Community Notes for many years now).
>
> [4] These are good questions!
> - Firstly, the factors in Figure 2 are not uniformly distributed in $[-1,1]$; as a crude estimate, they are closer to uniformly distributed in $[-0.66, 0.66]$. In general, it is harder to obtain extreme factors (close to $-1$ or $1$) due to the presence of regularization terms in the optimization objective (Eq. 2). We restrict our target to $[-0.5, 0.5]$ as it is attainable with relative ease (as we have demonstrated in Section 4) while still being sufficient to fabricate diverse agreement.
> - There might have been a typo in the question; we are not sure what you mean by $f_a$ and $f_{target}$. If you were referring to the attackers' factors, we would like to clarify that those terms are not learnt. We only learn note factors and intercepts.
> - We only consider 100 notes since this is already sufficient to obtain the spread of user factors sufficient for a successful attack.
> - Yes, choosing $i_{target}$ differently would affect the attack. In our experiments, we conservatively fix the target user intercept to be the population mean, but as discussed in Line 388 (column 2), our attack could potentially be more powerful if we also optimize for intercept choice; for instance, an adversary who strategically manipulates their intercept away from the mean (for instance, by cultivating a history of harsh ratings to achieve a low $i_u$) would gain additional leverage over the note intercept.
> - The method in Line 275 does not consider $0.5$ as a rating since it is always suboptimal to do so. In particular, since the change in intercept (Eq. 4) is an affine function of r (holding all else constant), it is always optimized at an extreme point (which is $0$ or $1$ in our case). We will add a clarification about this in our final manuscript to make this clear to the reader.
>
> We hope the reviewer finds our answers helpful. Please do not hesitate to let us know if any questions or concerns remain. Thank you again for your time and consideration.

---

> > ### Author Rebuttal · Reviewer_U3f1 · 2026-04-03
> >
> > Thank you for the responses. The paper is well motivated and interesting. I will keep my original score.

---

> > > ### Author Response · Authors · 2026-04-07
> > >
> > > Thank you for your response! We are glad to have addressed all your concerns and are happy to hear that you find our paper well motivated and interesting.

---

### Decision · Program_Chairs · 2026-04-30

**Decision:**

Accept (regular)

**Comment:**

This paper considers the robustness of crowdsourced fact-checking systems, focusing on algorithms used in Community Notes style mechanisms deployed by major platforms such as X, Meta, TikTok, and Google. These systems use a matrix-factorization based scoring algorithm. The paper show that this mechanism can be strategically manipulated by coordinated agents. By carefully choosing how they vote, attackers can artificially generate diverse agreement in the latent factor space. The paper proposes an attack strategy where a small coordinated group manipulates the consensus process. Experiments indicate that, in some cases, fewer than ten coordinated users may be sufficient to influence which notes are surfaced. Authors also discuss mitigation strategies for the attack: in particular, by proposing a cost model for the full attack and running a comparative estimate for the cost of manipulating votes and accounts.

Reviewers agree that the paper is well motivated, the topic is timely, and the limitations section is thoughtful. However, there is some debate regarding whether the paper provides sufficient contribution to the field. In particular, even after discussion, reviewers question the value of studying attacks on a relatively simple algorithm such as Community Notes. That said, as an initial study in this area, the work may offer enough value to merit broader visibility and discussion within the conference. Therefore, the paper is recommended for a weak accept. We wish the authors the best, and hope there will be room in the program for the paper to be accepted.